# META-LEARNING FOR STOCHASTIC GRADIENT MCMC

**Wenbo Gong**[1*], **Yingzhen Li**[2*†], **José Miguel Hernández-Lobato**[12]
[1]University of Cambridge   [2]Microsoft Research Cambridge
{wg242,jmh233}@cam.ac.uk, Yingzhen.Li@microsoft.com

## ABSTRACT

Stochastic gradient Markov chain Monte Carlo (SG-MCMC) has become increasingly popular for simulating posterior samples in large-scale Bayesian modeling. However, existing SG-MCMC schemes are not tailored to any specific probabilistic model, even a simple modification of the underlying dynamical system requires significant physical intuition. This paper presents the first meta-learning algorithm that allows automated design for the underlying continuous dynamics of an SG-MCMC sampler. The learned sampler generalizes Hamiltonian dynamics with state-dependent drift and diffusion, enabling fast traversal and efficient exploration of energy landscapes. Experiments validate the proposed approach on learning tasks with Bayesian fully connected neural networks, Bayesian convolutional neural networks and Bayesian recurrent neural networks, showing that the learned sampler out-performs generic, hand-designed SG-MCMC algorithms, and generalizes to different datasets and larger architectures.

## 1   INTRODUCTION

There is a resurgence of research interests in Bayesian deep learning (Graves, 2011; Blundell et al., 2015; Hernández-Lobato & Adams, 2015; Hernandez-Lobato et al., 2016; Gal & Ghahramani, 2016; Ritter et al., 2018), which applies Bayesian inference to neural networks for better uncertainty estimation. It is crucial for e.g. better exploration in reinforcement learning (Deisenroth & Rasmussen, 2011; Depeweg et al., 2017), resisting adversarial attacks (Feinman et al., 2017; Li & Gal, 2017; Louizos & Welling, 2017) and continual learning (Nguyen et al., 2018). A popular approach to performing Bayesian inference on neural networks is stochastic gradient Markov chain Monte Carlo (SG-MCMC), which adds properly scaled Gaussian noise to a stochastic gradient ascent procedure (Welling & Teh, 2011). Recent advances in this area further introduced optimization techniques such as pre-conditioning (Ahn et al., 2012; Patterson & Teh, 2013), annealing (Ding et al., 2014) and adaptive learning rates (Li et al., 2016a; Chen et al., 2016). All these efforts have made SG-MCMC highly scalable to many deep learning tasks, including shape and texture modeling in computer vision (Li et al., 2016b) and language modeling with recurrent neural networks (Gan et al., 2017). However, inventing novel dynamics for SG-MCMC requires significant mathematical work to ensure the sampler's stationary distribution is the target distribution, which is less friendly to practitioners. Furthermore, many of these algorithms are designed as a generic sampling procedure, and the associated physical mechanism might not be best suited for sampling neural network weights.

This paper aims to automate the SG-MCMC proposal design by introducing *meta-learning* techniques (Schmidhuber, 1987; Bengio et al., 1992; Naik & Mammone, 1992; Thrun & Pratt, 1998). The general idea is to train a *learner* on one or multiple tasks in order to acquire common knowledge that generalizes to future tasks. Recent applications of meta-learning include learning to transfer knowledge to unseen few-shot learning tasks (Santoro et al., 2016; Ravi & Larochelle, 2017; Finn et al., 2017), and learning algorithms such as gradient descent (Andrychowicz et al., 2016; Li & Malik, 2017; Wichrowska et al., 2017), Bayesian optimization (Chen et al., 2017) and reinforcement learning (Duan et al., 2016; Wang et al., 2016). Unfortunately, these advances cannot be directly transferred to the world of MCMC samplers, as a naive neural network parameterization of the transition kernel does not guarantee the posterior distribution to be the stationary distribution of the sampler.

---

[*]Equal contribution
[†]Work done at the University of Cambridge

We present to the best of our knowledge the first attempt towards meta-learning an SG-MCMC algorithm. Concretely, our contribution includes:

- An SG-MCMC sampler that extends Hamiltonian dynamics with *learnable* diffusion and curl matrices. Once trained, the sampler can generalize to different datasets and architectures.
- Extensive evaluation of the proposed sampler on Bayesian fully connected neural networks, Bayesian convolutional neural networks and Bayesian recurrent neural networks, with comparisons to popular SG-MCMC schemes based on e.g. Hamiltonian Monte Carlo (Chen et al., 2014) and pre-conditioned Langevin dynamics (Li et al., 2016a).

## 2 BACKGROUND: A COMPLETE FRAMEWORK FOR SG-MCMC

Consider sampling from a target density $\pi(\boldsymbol{\theta})$ that is defined by an *energy function*: $U(\boldsymbol{\theta}), \boldsymbol{\theta} \in \mathbb{R}^D$, $\pi(\boldsymbol{\theta}) \propto \exp(-U(\boldsymbol{\theta}))$. In this paper, we focus on this sampling task with a Bayesian modeling set-up, i.e. given observed data $\mathcal{D} = \{\boldsymbol{o}_n\}_{n=1}^N$, we define a probabilistic model $p(\mathcal{D}, \boldsymbol{\theta}) = \prod_{n=1}^N p(\boldsymbol{o}_n|\boldsymbol{\theta})p(\boldsymbol{\theta})$, and we want samples from the target density defined as *posterior distribution* $\pi(\boldsymbol{\theta}) = p(\boldsymbol{\theta}|\mathcal{D})$. We use Bayesian neural networks as an illustrating example, in this case, $\boldsymbol{o}_n = (\boldsymbol{x}_n, \boldsymbol{y}_n)$, the prior $p(\boldsymbol{\theta})$ is a Gaussian $\mathcal{N}(\boldsymbol{\theta}; \boldsymbol{0}, \lambda^{-1}\boldsymbol{I})$, and the energy function is defined as

$$U(\boldsymbol{\theta}) = -\sum_{n=1}^N \log p(\boldsymbol{y}_n|\boldsymbol{x}_n, \boldsymbol{\theta}) - \log p(\boldsymbol{\theta}) = \sum_{n=1}^N \ell(\boldsymbol{y}_n, \mathrm{NN}_{\boldsymbol{\theta}}(\boldsymbol{x}_n)) + \lambda||\boldsymbol{\theta}||_2^2, \qquad (1)$$

with $\ell(\boldsymbol{y}, \hat{\boldsymbol{y}})$ usually defined as the $\ell_2$ loss for regression or the cross-entropy loss for classification. A typical MCMC sampler constructs a Markov chain with a *transition kernel*, and corrects the proposed samples with Metropolis-Hastings (MH) rejection steps. Some of these methods, e.g. Hamiltonian Monte Carlo (HMC) (Duane et al., 1987; Neal et al., 2011), further augment the state space as $\boldsymbol{z} = (\boldsymbol{\theta}, \boldsymbol{r})$ with auxiliary variables $\boldsymbol{r}$, and sample from the augmented distribution $\pi(\boldsymbol{z}) \propto \exp(-H(\boldsymbol{z}))$, with the *Hamiltonian* $H(\boldsymbol{z}) = U(\boldsymbol{\theta}) + g(\boldsymbol{r})$ such that $\int \exp(-g(\boldsymbol{r}))d\boldsymbol{r} = C$. Thus, marginalizing out the auxiliary variable $\boldsymbol{r}$ will not affect the stationary distribution $\pi(\boldsymbol{\theta}) \propto \exp(-U(\boldsymbol{\theta}))$.

For deep learning tasks, the observed dataset $\mathcal{D}$ often contains thousands, if not millions, of instances, making MH rejection steps computationally prohibitive. Fortunately this is mitigated by SG-MCMC, whose transition kernel is implicitly defined by a stochastic differential equation (SDE) that leaves the target density invariant (Welling & Teh, 2011; Ahn et al., 2012; Patterson & Teh, 2013; Chen et al., 2014; Ding et al., 2014). Such a Markov process is called *Itô diffusion* governed by the continuous-time SDEs:

$$d\boldsymbol{z} = \boldsymbol{f}(\boldsymbol{z})dt + \sqrt{2\boldsymbol{D}(\boldsymbol{z})}d\boldsymbol{W}(t), \qquad (2)$$

with $\boldsymbol{f}(\boldsymbol{z})$ the deterministic *drift*, $\boldsymbol{W}(t)$ the Wiener process, and $\boldsymbol{D}(\boldsymbol{z})$ the *diffusion* matrix. As a simple example, Langevin dynamics considers $\boldsymbol{z} = \boldsymbol{\theta}$, $\boldsymbol{f}(\boldsymbol{\theta}) = -\nabla_{\boldsymbol{\theta}}U(\boldsymbol{\theta})$ and $\boldsymbol{D}(\boldsymbol{\theta}) = \mathbf{I}$. Then using *forward Euler discretization* with step-size $\eta$ the update rule of the parameters is

$$\boldsymbol{\theta}_{t+1} = \boldsymbol{\theta}_t - \eta \nabla_{\boldsymbol{\theta}_t} U(\boldsymbol{\theta}_t) + \sqrt{2\eta}\boldsymbol{\epsilon}_t, \quad \boldsymbol{\epsilon}_t \sim \mathcal{N}(\boldsymbol{0}, \mathbf{I}). \qquad (3)$$

Stochastic gradient Langevin dynamics (SGLD, Welling & Teh, 2011) proposed an approximation to (3), by replacing the exact gradient $\nabla_{\boldsymbol{\theta}}U(\boldsymbol{\theta})$ with an estimate using a mini-batch of datapoints:

$$\nabla_{\boldsymbol{\theta}}\tilde{U}(\boldsymbol{\theta}) = -\frac{N}{M}\sum_{m=1}^M \nabla_{\boldsymbol{\theta}} \log p(\boldsymbol{o}_m|\boldsymbol{\theta}) - \nabla_{\boldsymbol{\theta}} \log p(\boldsymbol{\theta}), \quad \boldsymbol{o}_1, ..., \boldsymbol{o}_M \sim \mathcal{D}. \qquad (4)$$

Therefore SGLD can be viewed as a stochastic gradient descent (SGD) that adds in a properly scaled Gaussian noise term. Similarly, SGHMC (Chen et al., 2014) is closely related to momentum SGD (see appendix). Furthermore, MH rejection steps are usually dropped in SG-MCMC when a carefully selected discretization step-size is in use. Therefore SG-MCMC has the same computational complexity as many stochastic optimization algorithms, making it highly scalable for sampling posterior distributions of neural network weights conditioned on big datasets.

Ma et al. (2015) derived a framework of SG-MCMC samplers using advanced statistical mechanics (Yin & Ao, 2006; Shi et al., 2012), which explicitly parameterizes the drift $\boldsymbol{f}(\boldsymbol{z})$ :

$$\boldsymbol{f}(\boldsymbol{z}) = -[\boldsymbol{D}(\boldsymbol{z}) + \boldsymbol{Q}(\boldsymbol{z})]\nabla_{\boldsymbol{z}}H(\boldsymbol{z}) + \boldsymbol{\Gamma}(\boldsymbol{z}), \quad \boldsymbol{\Gamma}_i(\boldsymbol{z}) = \sum_{j=1}^d \frac{\partial}{\partial \boldsymbol{z}_j}(\boldsymbol{D}_{ij}(\boldsymbol{z}) + \boldsymbol{Q}_{ij}(\boldsymbol{z})), \qquad (5)$$

with $\boldsymbol{Q}(\boldsymbol{z})$ the *curl* matrix, $\boldsymbol{D}(\boldsymbol{z})$ the diffusion matrix and $\boldsymbol{\Gamma}(\boldsymbol{z})$ a correction term. Remarkably Ma et al. (2015) showed the *completeness* of their framework:

1. $\pi(\boldsymbol{z}) \propto \exp(-H(\boldsymbol{z}))$ is a stationary distribution of the SDE (2)+(5) for any pair of positive semi-definite matrix $\boldsymbol{D}(\boldsymbol{z})$ and skew-symmetric matrix $\boldsymbol{Q}(\boldsymbol{z})$;

2. for any Itô diffusion process that has the unique stationary distribution $\pi(\boldsymbol{z})$, under mild conditions there exist $\boldsymbol{D}(\boldsymbol{z})$ and $\boldsymbol{Q}(\boldsymbol{z})$ matrices such that the process is governed by (2)+(5).

As a consequence, the construction of an SG-MCMC algorithm reduces to defining the state-space $\boldsymbol{z}$ and the $\boldsymbol{D}(\boldsymbol{z})$, $\boldsymbol{Q}(\boldsymbol{z})$ matrices. Indeed Ma et al. (2015) also cast existing SG-MCMC samplers within the framework, and proposed an improved version of SG-Riemannian-HMC. In general, an appropriate design of these two matrices leads to significant improvements in mixing as well as reduction of sample bias (Li et al., 2016a; Ma et al., 2015). However, this design has been historically based on strong physical intuitions from statistical mechanics (Duane et al., 1987; Neal et al., 2011; Ding et al., 2014). Therefore, it can still be difficult for practitioners to understand and engineer the sampling method that is best suited to their machine learning tasks.

In the next section, we will describe our recipe on meta-learning an SG-MCMC sampler of the form (2)+(5). Before the presentation, we emphasize that the *completeness* result of the framework is beneficial for our meta-learning task. On the one hand, as meta-learning searches the best algorithm for a given set of tasks, it is crucial that the search space is large enough to contain many useful candidates. On the other hand, some form of "correctness" guarantee is often required to achieve better generalization to test tasks that might not be very similar to the training tasks. Ma et al. (2015)'s completeness result indicates that our proposed method searches SG-MCMC samplers in the *biggest* subset of all Itô diffusion processes such that each instance is a *valid* posterior sampler. Therefore, the proposed meta-learning algorithm has the best from both worlds, indeed our experiments show that the learned sampler is superior to a number of other baseline SG-MCMC methods.

## 3 META-LEARNING FOR SG-MCMC

This section presents a meta-learning approach to learn an SG-MCMC proposal from data. Our aim is to design an appropriate parameterization of $\boldsymbol{D}(\boldsymbol{z})$ and $\boldsymbol{Q}(\boldsymbol{z})$, so that the sampler can be trained on simple tasks with a meta-learning procedure, and generalize to more complicated densities. For simplicity, we only augment the state-space by one extra variable $\boldsymbol{p}$ called *momentum* (Duane et al., 1987; Neal et al., 2011), although generalization to e.g. thermostat variable (Ding et al., 2014) is straightforward. Thus, the augmented state-space is $\boldsymbol{z} = (\boldsymbol{\theta}, \boldsymbol{p})$ (i.e. $\boldsymbol{r} = \boldsymbol{p}$), and the Hamiltonian is defined as $H(\boldsymbol{z}) = U(\boldsymbol{\theta}) + \frac{1}{2}\boldsymbol{p}^T\boldsymbol{p}$ with identity mass matrix.

### 3.1 EFFICIENT PARAMETERIZATION OF DIFFUSION AND CURL MATRICES

For neural networks, the dimensionality of $\boldsymbol{\theta}$ can be at least tens of thousands. Thus, training and applying full $\boldsymbol{D}(\boldsymbol{z})$ and $\boldsymbol{Q}(\boldsymbol{z})$ matrices can cause a huge computational burden, let alone gradient computations required by $\boldsymbol{\Gamma}(\boldsymbol{z})$. To address this, we define the preconditioning matrices as follows:

$$\boldsymbol{Q}(\boldsymbol{z}) = \begin{bmatrix} \boldsymbol{0} & -\boldsymbol{Q}_f(\boldsymbol{z}) \\ \boldsymbol{Q}_f(\boldsymbol{z}) & \boldsymbol{0} \end{bmatrix}, \quad \boldsymbol{D}(\boldsymbol{z}) = \begin{bmatrix} \boldsymbol{0} & \boldsymbol{0} \\ \boldsymbol{0} & \boldsymbol{D}_f(\boldsymbol{z}) \end{bmatrix},$$

$$\boldsymbol{Q}_f(\boldsymbol{z}) = \mathrm{diag}[\boldsymbol{f}_{\phi_Q}(\boldsymbol{z})], \quad \boldsymbol{D}_f(\boldsymbol{z}) = \mathrm{diag}[\alpha \boldsymbol{f}_{\phi_Q}(\boldsymbol{z}) \odot \boldsymbol{f}_{\phi_Q}(\boldsymbol{z}) + \boldsymbol{f}_{\phi_D}(\boldsymbol{z}) + c], \quad \alpha, c > 0. \tag{6}$$

Here $\boldsymbol{f}_{\phi_D}$ and $\boldsymbol{f}_{\phi_Q}$ are neural network parameterized functions that will be detailed in section 3.2, and $c$ is a small positive constant. We choose $\boldsymbol{D}_f$ and $\boldsymbol{Q}_f$ to be diagonal for fast computation, although future work can explore low-rank matrix solutions. From Ma et al. (2015), our design has the *unique* stationary distribution $\pi(\boldsymbol{\theta}) \propto \exp(-U(\boldsymbol{\theta}))$ if $\boldsymbol{f}_{\phi_D}$ is non-negative for all $\boldsymbol{z}$.

We discuss the role of each precondition matrix for better intuition. The curl matrix $\boldsymbol{Q}(\boldsymbol{z})$ in (2) mainly controls the deterministic drift forces introduced by the *energy gradient* $\nabla_{\boldsymbol{\theta}} U(\boldsymbol{\theta})$ (as seen in many HMC-like procedures and in eq. (5)). Usually, we only have access to the stochastic gradient $\nabla_{\boldsymbol{\theta}} \tilde{U}(\boldsymbol{\theta})$ through data sub-sampling, and here $\boldsymbol{D}(\boldsymbol{z})$ acts as the friction to counter for the associated noise that mainly affects the momentum $\boldsymbol{p}$. This explains the design of the diffusion matrix $\boldsymbol{D}(\boldsymbol{z})$ that uses $\boldsymbol{D}_f(\boldsymbol{z})$ to control the amount of friction and injected noise to the momentum. Furthermore,

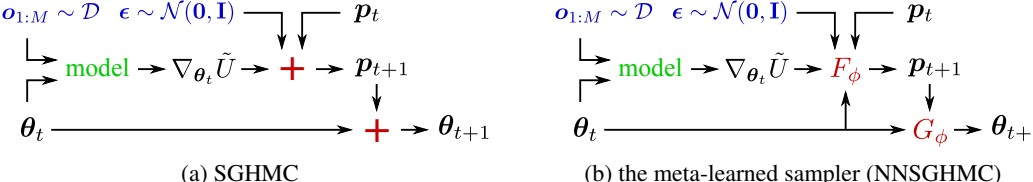

(a) SGHMC  (b) the meta-learned sampler (NNSGHMC)

Figure 1: Comparing the computation graphs of SGHMC and the meta-learned sampler (with modified forward Euler discretization). Here $F_\phi$ and $G_\phi$ transformations are defined by Eq. (7).

$D_f(z)$ should also account for the pre-conditioning effect introduced by $Q_f(z)$, e.g, when the magnitude of $Q_f(z)$ is large, we need higher friction correspondingly. This explains the squared term $f_{\phi_Q}(z) \odot f_{\phi_Q}(z)$ in $D_f(z)$ design. The positive scaling constant $\alpha$ is heuristically selected following (Chen et al., 2014; Ma et al., 2015) (see appendix). Finally, the extra term $\Gamma(z) = [\Gamma_\theta(z), \Gamma_p(z)]$ is responsible for compensating the changes introduced by preconditioning matrices $Q(z)$ and $D(z)$.

The discretized dynamics of the state $z = (\theta, p)$ with step-size $\eta$ and stochastic gradient $\nabla_\theta \tilde{U}(\theta)$ are

$$p_{t+1} = (1 - \eta D_f(z_t))p_t - \eta Q_f(z_t)\nabla_{\theta_t}\tilde{U}(\theta_t) + \eta\Gamma_p(z_t) + \epsilon, \quad \epsilon \sim \mathcal{N}(0, 2\eta D_f(z_t)),$$
$$\theta_{t+1} = \theta_t + \eta Q_f(\hat{z}_t)p_{t+1} + \eta\Gamma_\theta(\hat{z}_t), \qquad z_t = [\theta_t, p_t], \qquad \hat{z}_t = [\theta_t, p_{t+1}]. \tag{7}$$

We use a modified forward Euler discretization (Neal et al., 2011) here, and the computation graph of eq. (7) is visualized in the right part of Figure 1 (see appendix for SGHMC discretized updates). Again we see that $Q_f(z)$ is responsible for the acceleration of $\theta$, and from the $(1 - \eta D_f(z))$ term in the update equation of $p$, we see that $D_f(z)$ controls the friction introduced to the momentum. Note that in the big-data setting, the noisy gradient is approximately Gaussian distributed with mean $\mathbf{0}$ and variance $V(\theta)$. Observing this, Ma et al. (2015) further suggested a correction scheme to counter for stochastic gradient noise, which samples the Gaussian noise $\epsilon \sim \mathcal{N}(0, 2\eta D_f(z) - \eta^2\tilde{B}(\theta))$ with an empirical estimate of the variance $\tilde{B}(\theta) \approx Q_f(z)V(\theta)Q_f^T(z)$ instead. These corrections can be dropped when the discretization step-size $\eta$ is small, therefore, we do not consider them in our experiments.

## 3.2 CHOICES OF INPUTS TO THE NEURAL NETWORKS

We now present detailed functional forms for $f_{\phi_Q}$ and $f_{\phi_D}$. When designing these, our goal was to achieve a good balance between generalization power and computational efficiency. Recall that the curl matrix $Q(z)$ mainly controls the drift of the dynamics, and the desired behavior is the fast traverse through low-density regions. One useful source of information to identify this is the energy function $U(\theta)$.[1] We also include the momentum $p_i$ to the inputs of $f_{\phi_Q}$, allowing the $Q(z)$ matrix to observe the velocity information of the $\theta_i$. We further add an offset $\beta$ to $Q(z)$ to prevent the vanishing of this matrix. Putting all of them together, we define the $i^{\text{th}}$ element of $f_{\phi_Q}$ as

$$f_{\phi_Q,i}(z) = \beta + f_{\phi_Q}(U(\theta), p_i). \tag{8}$$

The corresponding $\Gamma(z)$ term requires both $\partial_{\theta_i} f_{\phi_Q}(U(\theta), p_i)$ and $\partial_{p_i} f_{\phi_Q}(U(\theta), p_i)$. The energy gradient $\partial_{\theta_i}\tilde{U}(\theta)$ also appears in (7), so it remains to compute $\partial_U f_{\phi_Q}$, which, along with $\partial_{p_i} f_{\phi_Q}(U(\theta), p_i)$, can be obtained by automatic differentiation (Abadi et al., 2015).

Matrix $D(z)$ is responsible for the friction and the stochastic gradient noise, which are crucial for better exploration around high-density regions. Therefore, we also add the energy gradient $\partial_{\theta_i} U(\theta)$ to the inputs, meaning that the $i^{\text{th}}$ element of $f_{\phi_D}$ is

$$f_{\phi_D,i}(z) = f_{\phi_D}(U(\theta), p_i, \partial_{\theta_i} U(\theta)). \tag{9}$$

By the construction of the $D(z)$ matrix, the $\Gamma$ vector only requires $\nabla_p D_f$ without computing any higher order information.

---

[1]The energy gradient $\nabla_\theta U(\theta)$ is also informative here, however, it requires expensive computation of the diagonal Hessian for $\Gamma(z)$. For similar reasons we do not consider other higher order derivatives as inputs.

In practice, both $U(\boldsymbol{\theta})$ and $\partial_{\theta_i} U(\boldsymbol{\theta})$ are replaced by their stochastic estimates $\tilde{U}(\boldsymbol{\theta})$ and $\partial_{\theta_i} \tilde{U}(\boldsymbol{\theta})$, respectively. To keep the scale of the inputs roughly the same across tasks, we rescale all the inputs using statistics computed by simulating the sampler with randomly initialized $\boldsymbol{f}_{\phi_D}$ and $\boldsymbol{f}_{\phi_Q}$. When the computational budget is limited, we replace the exact gradient computation required by $\boldsymbol{\Gamma}(\boldsymbol{z})$ with finite difference approximations. We refer the reader to the appendix for details.

### 3.3 Loss function design for meta-learning

Another challenge is to design a meta-learning procedure for the sampler to encourage faster convergence and low bias on test tasks. To achieve these goals we propose two loss functions that we named as the *cross-chain loss* and the *in-chain loss*. From now on we consider the discretized dynamics and define $q_t(\boldsymbol{\theta}|\mathcal{D})$ as the marginal distribution of the random variable $\boldsymbol{\theta}$ at time $t$.

**Cross-chain loss** We introduce *cross-chain loss* that encourages the sampler to converge faster. Since the sampler is guaranteed to have the unique stationary distribution $\pi(\boldsymbol{\theta}) \propto \exp(-U(\boldsymbol{\theta}))$, fast convergence means that $\mathrm{KL}[q_t || \pi]$ is close to zero when $t$ is small. Therefore this KL-divergence becomes a sensible objective to minimize, which is equivalent to maximizing the variational lower-bound (or ELBO): $\mathcal{L}_{\mathrm{VI}}^t(q_t) = -\mathbb{E}_{q_t}[U(\boldsymbol{\theta})] + \mathbb{H}[q_t]$ (Jordan et al., 1999; Beal, 2003). We further make the objective doubly stochastic: (1) the energy term is further approximated by its stochastic estimates $\tilde{U}(\boldsymbol{\theta})$; (2) we use Monte Carlo variational inference (MCVI, Ranganath et al., 2014; Blundell et al., 2015) which estimates the lower-bound with samples $\boldsymbol{\theta}_t^k \sim q_t(\boldsymbol{\theta}_t|\mathcal{D}), k = 1, ..., K$. These samples $\{\boldsymbol{\theta}_t^k\}_{k=1,t=1}^{K,T}$ are obtained by simulating $K$ parallel Markov chains with the sampler, and the cross-chain loss is defined by accumulating the lower-bounds through time:

$$\mathcal{L}_{\text{cross-chain}} = \frac{1}{T} \sum_{t=1}^{T} \mathcal{L}_{\mathrm{VI}}^t(\{\boldsymbol{\theta}_t^k\}_{k=1}^K), \quad \mathcal{L}_{\mathrm{VI}}^t(\{\boldsymbol{\theta}_t^k\}_{k=1}^K) = -\frac{1}{K} \sum_{k=1}^{K} \left[ \tilde{U}(\boldsymbol{\theta}_t^k) + \log q_t(\boldsymbol{\theta}_t^k|\mathcal{D}) \right]. \quad (10)$$

By minimizing this objective, we can improve the convergence of the sampler, especially at the early times of the Markov chain. The objective also takes the sampler bias into account because the two distributions will match when the KL-divergence is minimized.

**In-chain loss** For very big neural networks, simulating multiple Markov chains is prohibitively expensive. The issue is mitigated by *thinning* that collects samples for every $\tau$ step (after burn-in), which effectively draws samples from the averaged distribution $\bar{q}(\boldsymbol{\theta}|\mathcal{D}) = \frac{1}{\lfloor T/\tau \rfloor} \sum_{s=1}^{\lfloor T/\tau \rfloor} q_{s\tau}(\boldsymbol{\theta})$. The in-chain loss is, therefore, defined as the ELBO evaluated at the averaged distribution $\bar{q}$, which is then approximated by Monte Carlo with samples $\boldsymbol{\Theta}_{T,\tau}^k = \{\boldsymbol{\theta}_{s\tau}^k\}_{s=1}^{\lfloor T/\tau \rfloor}$ obtained by thinning:

$$\mathcal{L}_{\text{in-chain}} = \frac{1}{K} \sum_{k=1}^{K} \mathcal{L}_{\mathrm{VI}}^k\left(\boldsymbol{\Theta}_{T,\tau}^k\right), \quad \mathcal{L}_{\mathrm{VI}}^k\left(\boldsymbol{\Theta}_{T,\tau}^k\right) = -\frac{1}{\lfloor T/\tau \rfloor} \sum_{s=1}^{\lfloor T/\tau \rfloor} \left[ \tilde{U}(\boldsymbol{\theta}_{s\tau}^k) + \log \bar{q}(\boldsymbol{\theta}_{s\tau}^k|\mathcal{D}) \right]. \quad (11)$$

**Gradient approximation** We leverage the recently proposed Stein gradient estimator (Li & Turner, 2018) to estimate the intractable gradients $\nabla_\phi \log q_t(\boldsymbol{\theta})$ for cross-chain loss and $\nabla_\phi \log \bar{q}(\boldsymbol{\theta})$ for in-chain loss. Precisely, by the chain rule, we have $\nabla_\phi \log q_t(\boldsymbol{\theta}) = \nabla_\phi \boldsymbol{\theta} \nabla_{\boldsymbol{\theta}} \log q_t(\boldsymbol{\theta})$, so it remains to estimate the gradients $\boldsymbol{G} = (\nabla_{\boldsymbol{\theta}_t^1} \log q_t(\boldsymbol{\theta}_t^1), \ldots, \nabla_{\boldsymbol{\theta}_t^K} \log q_t(\boldsymbol{\theta}_t^K))^T$ at the sampled locations $\{\boldsymbol{\theta}_t^k\}_{k=1}^K \sim q_t$. The recipe first constructs a kernel matrix $\boldsymbol{K}$ with $\boldsymbol{K}_{ij} = \mathcal{K}(\boldsymbol{\theta}_t^i, \boldsymbol{\theta}_t^j)$, and then estimates the gradients by $\boldsymbol{G} \approx -(\boldsymbol{K} + \lambda \boldsymbol{I})^{-1} \langle \nabla, \boldsymbol{K} \rangle$, where $\langle \nabla, \boldsymbol{K} \rangle_{ij} = \sum_{k=1}^{K} \partial_{\boldsymbol{\theta}_t^k(j)} \mathcal{K}(\boldsymbol{\theta}_t^k, \boldsymbol{\theta}_t^i)$. In our experiments, we use the RBF kernel, and the corresponding gradient estimator has a simple analytic form that can be computed efficiently in $\mathcal{O}(K^2 D + K^3)$ time (usually $K \ll D$).

## 4 Related work

Since the development of SGLD (Welling & Teh, 2011), SG-MCMC has been increasingly popular for posterior sampling on big data. In detail, Chen et al. (2014) scaled up HMC with stochastic gradients, Ding et al. (2014) further augmented the state space with an auxiliary temperature variable, and Springenberg et al. (2016) improved robustness through scale adaptation. The SG-MCMC

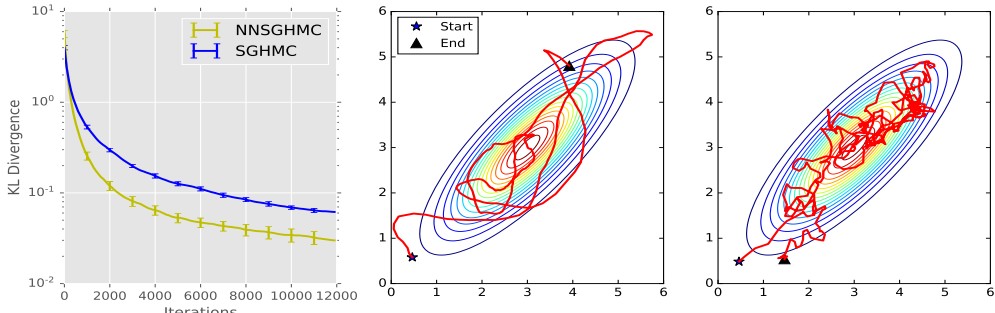

Figure 2: (Left) Sampler's bias measured by KL. (Middle) NNSGHMC trajectory plot on a 2D-Gaussian with manually injected gradient noise. (Right) SGHMC plot for the same settings.

extensions to Riemannian Langevin dynamics and HMC (Girolami & Calderhead, 2011) have also been proposed (Patterson & Teh, 2013; Ma et al., 2015). Our proposed sampler architecture further generalizes SG-Riemannian-HMC as it decouples the design of $D(z)$ and $Q(z)$ matrices, and the detailed functional form of these two matrices are also learned from data.

Our approach is closely related to the recent line of work on learning optimization algorithms. Specifically, Andrychowicz et al. (2016) trained a recurrent neural network (RNN) based optimizer that transfers to similar tasks with supervised learning. Later Chen et al. (2017) generalized this approach to Bayesian optimization (Brochu et al., 2010; Snoek et al., 2012) which is gradient-free. We do not use RNNs in our approach as it cannot be represented within the framework of Ma et al. (2015). We leave the combination of learnable RNN proposals to future work. Also Li & Turner (2018) presented an initial attempt to meta-learn an approximate inference algorithm, which simply combined the stochastic gradient and the Gaussian noise with a neural network. Thus the stationary distribution of that sampler (if it exists) is only an approximation to the exact posterior. On the other hand, the proposed sampler (with $\eta \to 0$) is guaranteed to be correct by the complete framework (Ma et al., 2015). Very recently Wu et al. (2018) discussed that short-horizon meta-objectives for learning optimizers can cause a serious issue for long-time generalization. We found this bias is less severe in our approach, again due to the fact that the learned sampler is provably correct.

Recent research also considered improving HMC with a trainable transition kernel. Salimans et al. (2015) improved upon vanilla HMC by introducing a trainable re-sampling distribution for the momentum. Song et al. (2017) parameterized the HMC transition kernel with a trainable invertible transformation called non-linear independent components estimation (NICE, Dinh et al., 2014), and train it with Wasserstein adversarial training (Arjovsky et al., 2017). Levy et al. (2018) generalized HMC by augmenting the state space with a binary direction variable, and they parameterized the transition kernel with a non-volume preserving invertible transformation that is inspired by the real-valued non-volume preserving (RealNVP) flows (Dinh et al., 2017). The sampler is trained with the expected squared jump distance (Pasarica & Gelman, 2010). We note that adversarial training is less reliable for high dimensional data, thus it is not considered in this paper. Also, the jump distance does not explicitly take the sampling bias and convergence speed into account. More importantly, the purpose of these approaches is to directly improve the HMC-like sampler on the *target distribution*, and with NICE/RealNVP parametrization it is difficult to generalize the sampler to densities of different dimensions. In contrast, our goal is to learn an SG-MCMC sampler that can later be transferred to sample from *different* Bayesian neural network posterior distributions, which will typically have *different* dimensionality and include tens of thousands of random variables.

## 5 EXPERIMENTS

We evaluate the meta-learned SG-MCMC sampler, which is referred to as NNSGHMC or the meta sampler in the following. Detailed test set-ups are reported in the appendix. The code is available at https://github.com/WenboGong/MetaSGMCMC.

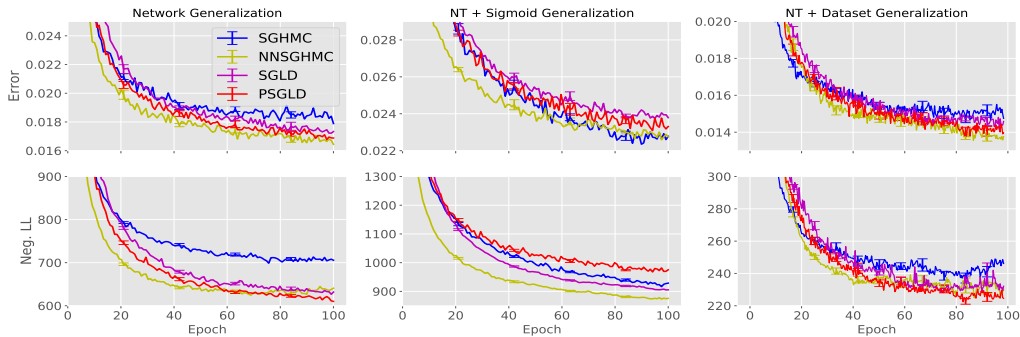

Figure 3: Learning curves on test error (top) and negative test LL (bottom).

Table 1: The final performance for the samplers on MNIST, averaged over 10 independent runs.

| Sampler | NT Err. | NT+AF Err | NT+Data Err | NT NLL | NT+AF NLL | NT+Data NLL |
|---|---|---|---|---|---|---|
| NNSGHMC | **98.36±0.02**% | **97.72±0.02**% | **98.62±0.02**% | 640±6.25 | **875±3.19** | 230±3.23 |
| SGHMC | 98.21±0.01% | **97.72±0.01**% | 98.52±0.03% | 705±3.44 | 929±2.95 | 246±5.43 |
| SGLD | 98.27±0.02% | 97.62±0.02% | 98.54±0.01% | 631±3.15 | 905±2.36 | 232±1.93 |
| PSGLD | 98.31±0.02% | 97.67±0.02% | 98.60±0.02% | **610±2.93** | 975±4.41 | **224±1.97** |

## 5.1 SYNTHETIC EXAMPLE: SAMPLING GAUSSIAN VARIABLES WITH NOISY GRADIENTS

We first consider sampling Gaussian variables to demonstrate fast convergence and low bias of the meta sampler. To mimic stochastic gradient settings, we manually inject Gaussian noise with unit variance to the gradient as suggested by (Chen et al., 2014). The training density is a 10D Gaussian with randomly generated diagonal covariance matrix, and the test density is a 20D Gaussian. For evaluation, we simulate $K = 50$ parallel chains for $T = 12,000$ steps. Then we follow Ma et al. (2015) to evaluate the sampler's bias measured by the KL divergence from the empirical estimate to the ground truth. Results are visualized on the left panel of Figure 2, showing that the meta sampler both converges much faster and achieves lower bias compared to SGHMC. The effective sample size[2] for SGHMC and NNSGHMC are **22** and **59**, again indicating better efficiency of the meta sampler. For illustration purposes, we also plot in the other two panels the trajectory of samples by simulating NNSGHMC (middle) and SGHMC (right) on a 2D Gaussian for a fixed amount of time $\eta T$. This confirms that the meta sampler explores more efficiently and is less affected by the injected noise.

## 5.2 BAYESIAN FEEDFORWARD NEURAL NETWORKS

Next, we consider Bayesian neural network classification on MNIST data with three generalization tests: *network architecture generalization* (NT), *activation function generalization* (AF) and *dataset generalization* (Data). In all tests, the sampler is trained with a 1-hidden layer multi-layer perceptron (MLP) (20 units, ReLU activation) as the underlying model for the target distribution $\pi(\boldsymbol{\theta})$. We also report long-time horizon generalization results, meaning that the simulation time steps in test time are much longer than that of training (cf. Andrychowicz et al., 2016). Algorithms in comparison include SGLD (Welling & Teh, 2011), SGHMC (Chen et al., 2014) and preconditioned SGLD (PSGLD, Li et al., 2016a). Note that PSGLD uses RMSprop-like preconditioning techniques (Tieleman & Hinton, 2012) that require moving average estimates of the gradient's second moments. Therefore the underlying dynamics of PSGLD cannot be represented within our framework (6). Thus we mainly focus on comparisons with SGLD and SGHMC, and leave the PSGLD results as reference. The discretization step-sizes for the samplers are tuned on the validation dataset for each task.

**Architecture generalization (NT)** In this test we use the trained sampler to draw samples from the posterior distribution of a *2-hidden layer* MLP with *40 units* and ReLU activations. Figure 3 shows the learning curves of test error and negative test log-likelihood (NLL) for 100 epochs, where the final performance is reported in Table 1. Overall NNSGHMC achieves the fastest convergence even when compared with PSGLD. It has the lowest test error compared to SGLD and SGHMC. NNSGHMC's final test LL is on par with SGLD and slightly worse than PSGLD, but it is still better than SGHMC.

---

[2]Implementation follows the ESS function in the BEAST package `http://beast.community`.

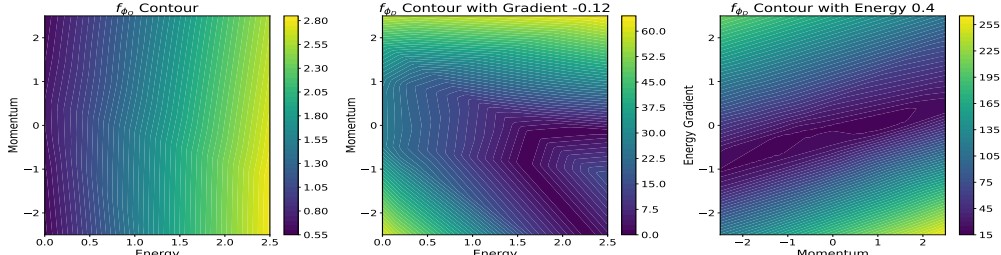

Figure 4: (Left) The contour plot of function $\boldsymbol{f}_{\phi_Q}$ (Middle) The contour plot for $\boldsymbol{f}_{\phi_D}$ for dimension 1 and 2 with fixed $-\nabla_{\boldsymbol{\theta}}U(\boldsymbol{\theta})$ (Right) The same plot for $\boldsymbol{f}_{\phi_D}$ for dimension 2 and 3 with fixed energy.

**Architecture + Activation function generalization (NT+AF)**   Next we replace NT's test network's activation function with **sigmoid** and re-run the same test as before. Again results in Figure 3 and Table 1 show that NNSGHMC converges faster than others for both test error and NLL. It also achieves the best NLL results among all samplers, and the same test error as SGHMC.

**Architecture + Dataset generalization (NT+Data)**   In this test we split MNIST into *training task* (classifying digits 0-4) and *test task* (digits 5-9). The meta sampler is trained with the smaller MLP, and it is evaluated on the task with the larger MLP with NT's architecture. Thus, the meta sampler is trained without any knowledge of the test task's training and test data. From Figure 3, we see that NNSGHMC, although a bit slower at the start, catches up quickly and proceeds to lower error. The difference between these samplers NLL results is marginal, and NNSGHMC is on par with PSGLD.

**Learned strategies**   For better intuition, we visualize in Figure 4 the contours of $\boldsymbol{f}_{\phi_D}$ and $\boldsymbol{f}_{\phi_Q}$. From the left panel, $\boldsymbol{f}_{\phi_Q}$ has learned a nearly linear strategy w.r.t. the energy and small variations w.r.t. the momentum. This enables the sampler for fast traversal through low density (high energy) regions and better exploration at high density (low energy) area.

The strategy learned for the diffusion matrix $\boldsymbol{D}(\boldsymbol{z})$ is rather interesting. Recall that $\boldsymbol{D}(\boldsymbol{z})$ is parametrized by both $\boldsymbol{f}_{\phi_D}$ and $\boldsymbol{f}_{\phi_Q} \odot \boldsymbol{f}_{\phi_Q}$ (eq. (6)). Since Figure 4 (left) indicates that $\boldsymbol{f}_{\phi_Q}$ is large in high energy regions, the amount of friction is adequate, so $\boldsymbol{f}_{\phi_D}$ tends to reduce its output to maintain momentum (see the middle plot). By contrast, at low energy regions, $\boldsymbol{f}_{\phi_D}$ increases to add friction in order to prevent divergence. The right panel visualizes the interactions between the momentum and the mean gradient $-\frac{1}{N}\nabla_{\boldsymbol{\theta}}U(\boldsymbol{\theta})$ at a fixed energy level. This indicates that the meta sampler has learned a strategy to prevent overshoot by producing large friction, indeed $\boldsymbol{f}_{\phi_D}$ returns large values when the signs of the momentum and the gradient differ.

### 5.3  BAYESIAN CONVOLUTIONAL NEURAL NETWORKS

Following the setup of BNN MNIST experiments, we also test our algorithm on convolutional neural networks (CNNs) for CIFAR-10 (Krizhevsky, 2009) classification, again with three generalization tasks (NT, AF and Data). The meta sampler is trained using a smaller CNN with two convolutional layers ($3 \times 3 \times 3 \times 8$ and $3 \times 3 \times 8 \times 8$) and one fully connected (fc) layer (50 hidden units). ReLU activations, and max-pooling operators of size 2 are applied after each convolutional layer. The meta sampler is trained using 100 "meta-epochs", where each "meta-epoch" has 5 data epochs. At the beginning of each "meta-epoch", a "replay" technique inspired by experience replay (Lin, 1993) is utilized (see appendix). The discretization step-sizes are tuned on a validation dataset for each task.

**Architecture generalization (NT)**   The test CNN has two convolutional layers ($3 \times 3 \times 3 \times 16$ and $3 \times 3 \times 16 \times 16$) and one fc layer (100 hidden units), resulting in roughly $4\times$ dimenality of $\boldsymbol{\theta}$. Figure 5 shows that the meta sampler achieves the fastest learning at the first 10 epochs, and continues to have better performance in both test accuracy and NLL. Interestingly, PSGLD slows down quickly after 3 epochs, and it converges to a worse answer. The best performance over 200 epochs is shown in Table 2, where the meta sampler is a clear winner in both accuracy and NLL. This demonstrates that our sampler indeed converges faster and has found a better posterior mode.

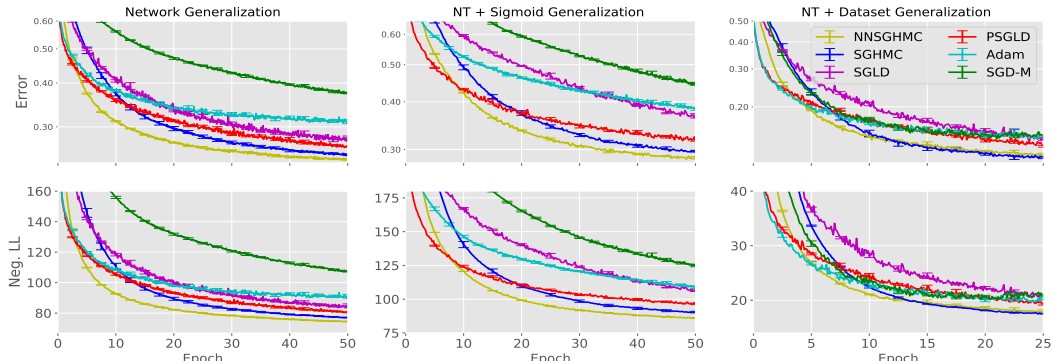

Figure 5: Learning curves on test error (top) and negative test LL/100 (bottom).

Table 2: The best performance on CIFAR-10 over 200 epoch , averaged over 5 independent runs. All the samplers achieved the best performance after around 190 epochs.

| Methods | NT Err. | NT+AF Err | NT+Data Err | NT NLL/100 | NT+AF NLL/100 | NT+Data NLL/100 |
|---|---|---|---|---|---|---|
| NNSGHMC | **78.12**±0.035% | **74.41**±**0.11**% | 89.97±0.04% | **68.88**±0.15 | **79.55**±**0.057** | 15.66±0.28 |
| SGHMC | 77.63±0.068% | 73.68±0.17% | **90.11**±**0.15**% | 70.39±0.27 | 82.08±0.48 | **15.26**±0.07 |
| SGLD | 76.38±0.085% | 73.83±0.013% | 89.34±0.06% | 73.39±0.26 | 80.30±0.14 | 16.36±0.07 |
| PSGLD | 77.46±0.05% | 73.16±0.1% | 89.78±0.08% | 69.89±0.2 | 83.70±0.21 | 15.72±0.04 |
| Adam | 70.94±0.10% | 69.73±0.11% | 85.44±0.14% | 86.77±1.01 | 87.60±0.44 | 22.35±0.47 |
| SGD-M | 68.06±0.27% | 68.76±0.17% | 84.86±0.48% | 99.12±1.06 | 90.35±0.29 | 23.64±0.73 |

**Architecture + Activation function generalization (NT+AF)** We use the same CNN architecture as in NT but replace the ReLU activations with sigmoid. Figure 5 and Table 2 show that the meta sampler again has better convergence speed and the best final performance.

**Architecture + Dataset generalization (NT+Data)** We split CIFAR-10 according to labels 0-4 as the training task and 5-9 as the test task. We also used the same CNN architecture as in NT. From Figure 5 and Table 2, the meta sampler consistently achieves the fastest convergence speed. It also achieves similar accuracy as SGHMC, but it has slightly worse test NLL compared to SGHMC.

## 5.4 BAYESIAN RECURRENT NEURAL NETWORKS

Lastly, we consider a more challenging setup: sequence modeling with Bayesian RNNs. Here a single datum is a sequence $\boldsymbol{o}_n = \{\boldsymbol{x}_n^1, ..., \boldsymbol{x}_n^T\}$ and the log-likelihood is defined as $\log p(\boldsymbol{o}_n|\boldsymbol{\theta}) = \sum_{t=1}^{T} \log p(\boldsymbol{x}_t^n|\boldsymbol{x}_1^n, \ldots, \boldsymbol{x}_{t-1}^n, \boldsymbol{\theta})$, with each of the conditional densities produced by a gated recurrent unit (GRU) network (Cho et al., 2014). We consider four polyphonic music datasets for this task: Piano-midi (Piano) as training data, and Nottingham (Nott), MuseData (Muse) and JSB chorales (JSB) for evaluation. The meta sampler is trained on a small GRU with 100 hidden states. At test time, we follow Chen et al. (2016) and set the step-size to $\eta = 0.001$. We found SGLD significantly under-performs, so instead, we report the performances of two optimizers, Adam (Kingma & Ba, 2014) and Santa (Chen et al., 2016), taken from Chen et al. (2016). Again, these two optimizers use moving average schemes which are out of the scope of our framework, so we mainly compare the meta sampler with SGHMC and leave the others as references.

The meta sampler is tested on the four datasets using 200 unit GRU. So for Piano this corresponds to architecture generalization only. From Figure 6 we see that the meta sampler achieves faster convergence compared to SGHMC, at the same time it achieves similar speed as Santa at early stages. All the samplers achieve best results close to Santa on Piano. The meta sampler successfully generalizes to the other three datasets, demonstrating faster convergence than SGHMC consistently, and better final performance on Muse. Interestingly, the meta sampler's final results on Nott and JSB are slightly worse than other samplers. Presumably, these two datasets are very different from Muse and Piano, therefore, the energy landscape is less similar to the training density (see appendix). Specifically, JSB is a dataset with much shorter sequences. And in this case, SGHMC also exhibits over-fitting but to a smaller degree. Therefore, we further test the meta sampler on JSB without the offset $\beta$ in $\boldsymbol{f}_{\phi_Q}$ to reduce the acceleration (denoted as NNSGHMC-s). Surprisingly, NNSGHMC-s

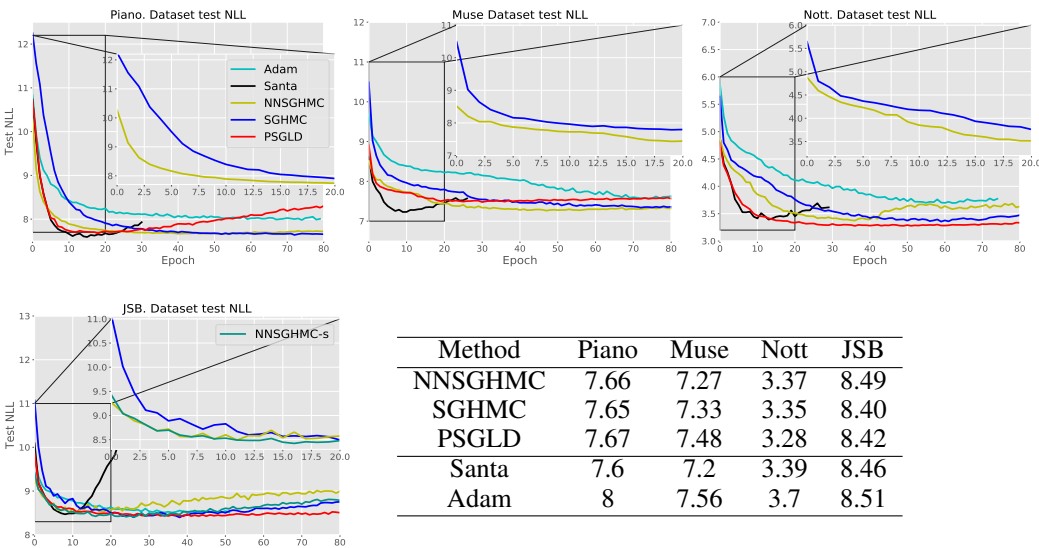

Figure 6: Test NLL learning curve (with zoom-in for sampling methods) and the best performance. Santa and Adam results are from Chen et al. (2016)

| Method | Piano | Muse | Nott | JSB |
|--------|-------|------|------|-----|
| NNSGHMC | 7.66 | 7.27 | 3.37 | 8.49 |
| SGHMC | 7.65 | 7.33 | 3.35 | 8.40 |
| PSGLD | 7.67 | 7.48 | 3.28 | 8.42 |
| Santa | 7.6 | 7.2 | 3.39 | 8.46 |
| Adam | 8 | 7.56 | 3.7 | 8.51 |

convergences in similar speeds as the original one, but with less amount of over-fitting and better final test NLL **8.40**.

## 6 CONCLUSIONS AND FUTURE WORK

We have presented a meta-learning algorithm that can learn an SG-MCMC sampler on simpler tasks and generalizes to more complicated densities in high dimensions. Experiments on Bayesian MLPs, Bayesian CNNs and Bayesian RNNs confirmed the strong generalization of the trained sampler to the long-time horizon as well as across datasets and network architectures. Future work will focus on better designs for both the sampler and the meta-learning procedure. For the former, temperature variable augmentation as well as moving average estimation will be explored. For the latter, better loss functions will be proposed for faster training, e.g. by reducing the unrolling steps of the sampler during training. Finally, the automated design of generic MCMC algorithms that might not be derived from continuous Markov processes remains an open challenge.

### ACKNOWLEDGMENTS

We thank Shixiang Gu, Mark Rowland and Cheng Zhang for comments on the manuscript. We also appreciate Changyou Chen for providing the experiment results of Bayesian RNN (Chen et al., 2016). Wenbo Gong is supported by the CSC-Cambridge Trust Scholarship.

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

## A    COMPARING MOMENTUM SGD AND SGHMC

Similar to the relationship between SGLD and SGD, SGHMC is closely related SGD with momentum (SGD-M). First in HMC, the state space is augmented with an additional momentum variable denoted as $\boldsymbol{p} \in \mathbb{R}^D$. We assume an identity mass matrix associated with that momentum term. Then the corresponding drift $\boldsymbol{f}(\boldsymbol{\theta}, \boldsymbol{p})$ and diffusion matrix $\boldsymbol{D}$ are:

$$\boldsymbol{f}(\boldsymbol{\theta}, \boldsymbol{p}) = \begin{bmatrix} \mathbf{0} & \boldsymbol{I} \\ -\boldsymbol{I} & -\boldsymbol{C} \end{bmatrix} \begin{bmatrix} \nabla U(\boldsymbol{\theta}) \\ \boldsymbol{p} \end{bmatrix}, \qquad \boldsymbol{D} = \begin{bmatrix} \mathbf{0} & \mathbf{0} \\ \mathbf{0} & \boldsymbol{C} \end{bmatrix}, \tag{12}$$

where $\boldsymbol{C}$ is a positive definite matrix called *friction* coefficient. Thus, HMC's continuous-time dynamics is governed by the following SDE:

$$\begin{aligned} d\boldsymbol{\theta} &= \boldsymbol{p}dt, \\ d\boldsymbol{p} &= -\nabla U(\boldsymbol{\theta})dt - \boldsymbol{C}\boldsymbol{p}dt + \sqrt{2\boldsymbol{C}}d\boldsymbol{W}(t). \end{aligned} \tag{13}$$

The discretized update rule (with simple Euler discretization) of HMC with step-size $\eta$ is

$$\begin{aligned} \boldsymbol{\theta}_{t+1} &= \boldsymbol{\theta}_t + \eta\boldsymbol{p}_t, \\ \boldsymbol{p}_{t+1} &= (1 - \eta\boldsymbol{C})\boldsymbol{p}_t - \eta\nabla U(\boldsymbol{\theta}_t) + \boldsymbol{\epsilon}_t, \\ \boldsymbol{\epsilon}_t &\sim \mathcal{N}(\mathbf{0}, 2\eta\boldsymbol{C}). \end{aligned} \tag{14}$$

If stochastic gradient $\nabla \tilde{U}(\boldsymbol{\theta})$ is used, we need to replace the covariance matrix of $\boldsymbol{\epsilon}$ with $2\eta(\boldsymbol{C} - \hat{\boldsymbol{B}})$ where $\hat{\boldsymbol{B}}$ is the variance estimation of the gradients.

On the other hand, the update equations of SGD with momentum (SGD-M) are the following:

$$\begin{aligned} \boldsymbol{\theta}_{t+1} &= \boldsymbol{\theta}_t + \boldsymbol{v}_t, \\ \boldsymbol{v}_{t+1} &= k\boldsymbol{v}_t - l\nabla U(\boldsymbol{\theta}_t). \end{aligned} \tag{15}$$

where $k$ and $l$ are called *momentum discount factor* and *learning rate*, respectively. Also we can rewrite the SGHMC update equations by setting $\eta\boldsymbol{p}_t = \boldsymbol{v}_t$, $l = \eta^2$, $k = (1 - \eta\boldsymbol{C})$,

$$\begin{aligned} \boldsymbol{\theta}_{t+1} &= \boldsymbol{\theta}_t + \boldsymbol{v}_t, \\ \boldsymbol{v}_{t+1} &= k\boldsymbol{v}_t - l\nabla U(\boldsymbol{\theta}_t) + \hat{\boldsymbol{\epsilon}}_t, \\ \hat{\boldsymbol{\epsilon}}_t &\sim \mathcal{N}(\mathbf{0}, 2l(1 - k)). \end{aligned} \tag{16}$$

Thus, the discretized SGHMC updates can be viewed as the SGD-M update injected with carefully controlled Gaussian noise. Therefore, the hyperparameter of SGHMC can be heuristically chosen based on the experience of SGD-M and vice versa.

Neal et al. (2011) showed that in practice, simple Euler discretization for HMC simulation might cause divergence, therefore advanced discretization schemes such as Leapfrog and modified Euler are recommended. We use modified Euler discretization in our implementation of SGHMC and the meta sampler, resulting in the following update:

$$\begin{aligned} \boldsymbol{p}_{t+1} &= (1 - \eta\boldsymbol{C})\boldsymbol{p}_t - \eta\nabla U(\boldsymbol{\theta}_t) + \boldsymbol{\epsilon}_t, \\ \boldsymbol{\theta}_{t+1} &= \boldsymbol{\theta}_t + \eta\boldsymbol{p}_{t+1}, \\ \boldsymbol{\epsilon}_t &\sim \mathcal{N}(\mathbf{0}, 2\eta\boldsymbol{C}). \end{aligned} \tag{17}$$

## B    FINITE DIFFERENCE APPROXIMATION FOR THE GAMMA VECTOR

The main computational burden is the gradient computation required by $\boldsymbol{\Gamma}(\boldsymbol{z})$ vector. To simplify notations we will write $\boldsymbol{D} = \boldsymbol{D}(\boldsymbol{z})$ and $\boldsymbol{Q} = \boldsymbol{Q}(\boldsymbol{z})$. From the parametrization of the $\boldsymbol{Q}$ and $\boldsymbol{D}$ matrices in eq. (6), for $\boldsymbol{\theta}, \boldsymbol{p} \in \mathbb{R}^D$ we have $\boldsymbol{\Gamma}(\boldsymbol{z}) = [\boldsymbol{\Gamma_\theta}, \boldsymbol{\Gamma_p}]$. For the first term $\boldsymbol{\Gamma_\theta}$, we have

$$\boldsymbol{\Gamma}_{\boldsymbol{\theta},i} = -\nabla_{\boldsymbol{p}} \cdot \boldsymbol{Q}_{i,:} = -\frac{\partial f_{\phi_Q, i}}{\partial p_i}. \tag{18}$$

Due to the two-stage update of Euler integrator, at time t, we have $f_{\phi_Q, i}^{t-1} = f_{\phi_Q, i}(U(\boldsymbol{\theta}_{t-1}), p_i^{t-1})$, $\hat{f}_{\phi_Q, i}^{t-1} = f_{\phi_Q, i}(U(\boldsymbol{\theta}_{t-1}), p_i^t)$ and $f_{\phi_D, i}^{t-1} = f_{\phi_D, i}(U(\boldsymbol{\theta}_{t-1}), p_i^{t-1}, \nabla_{\theta_i^{t-1}} U(\boldsymbol{\theta}_{t-1}))$. Thus a proper finite

difference method requires $f_{\phi_Q,i}(U(\boldsymbol{\theta}_t), p_i^{t-1})$, which is not exactly the history from the previous time. Therefore we further approximate it using delayed estimate:

$$\frac{\partial f_{\phi_Q,i}^t}{\partial p_i^t} \approx \frac{\hat{f}_{\phi_Q,i}^{t-1} - f_{\phi_Q,i}^{t-1}}{p_i^t - p_i^{t-1}} \quad \Rightarrow \quad \boldsymbol{\Gamma}_{\boldsymbol{\theta}}^t \approx -\frac{\hat{\boldsymbol{Q}}^{t-1} - \boldsymbol{Q}^{t-1}}{\boldsymbol{p}^t - \boldsymbol{p}^{t-1}}. \tag{19}$$

Similarly, the $\boldsymbol{\Gamma_p}$ term expands as

$$\begin{aligned}
\boldsymbol{\Gamma}_{\boldsymbol{p},i} &= \nabla \cdot [\boldsymbol{D} + \boldsymbol{Q}]_{i,:} \\
&= \frac{\partial f_{\phi_Q,i}}{\partial \theta_i} + \frac{\partial f_{\phi_D,i}}{\partial p_i} + 2\alpha f_{\phi_Q,i} \frac{\partial f_{\phi_Q,i}}{\partial p_i} \\
&= \frac{\partial f_{\phi_Q,i}}{\partial U(\boldsymbol{\theta})} \frac{\partial U(\boldsymbol{\theta})}{\partial \theta_i} + \frac{\partial f_{\phi_D,i}}{\partial p_i} + 2\alpha f_{\phi_Q,i} \frac{\partial f_{\phi_Q,i}}{\partial p_i}.
\end{aligned} \tag{20}$$

We further approximate $\frac{\partial f_{\phi_Q,i}}{\partial U(\boldsymbol{\theta})}$ by the following

$$\frac{\partial f_{\phi_Q,i}}{\partial U(\boldsymbol{\theta})} \approx \frac{f_{\phi_Q,i}^t - \hat{f}_{\phi_Q,i}^{t-1}}{U(\boldsymbol{\theta}_t) - U(\boldsymbol{\theta}_{t-1})} \tag{21}$$

This only requires the storage of previous $\boldsymbol{Q}$ matrix. However, $\frac{\partial f_{\phi_D,i}}{\partial p_i}$ requires one further forward pass to obtain $\hat{f}_{\phi_D,i}^{t-1} = f_{\phi_D,i}(U(\boldsymbol{\theta}_t), p_i^{t-1}, \nabla_{\theta_i^t} U(\boldsymbol{\theta}_t))$, thus, we have

$$\begin{aligned}
\frac{\partial f_{\phi_D,i}}{\partial p_i} &\approx \frac{f_{\phi_D,i}^t - \hat{f}_{\phi_D,i}^{t-1}}{p_i^t - p_i^{t-1}} \\
\Rightarrow \boldsymbol{\Gamma_p} &\approx \frac{\boldsymbol{Q}^t - \hat{\boldsymbol{Q}}^{t-1}}{U(\boldsymbol{\theta}_t) - U(\boldsymbol{\theta}_{t-1})} \odot \nabla_{\boldsymbol{\theta}} U(\boldsymbol{\theta}_t) + \frac{\boldsymbol{f}_{\phi_D}^t - \hat{\boldsymbol{f}}_{\phi_D}^{t-1}}{\boldsymbol{p}^t - \boldsymbol{p}^{t-1}} + 2\alpha \boldsymbol{f}_{\phi_Q}^t \frac{\hat{\boldsymbol{Q}}^{t-1} - \boldsymbol{Q}^{t-1}}{\boldsymbol{p}^t - \boldsymbol{p}^{t-1}}.
\end{aligned} \tag{22}$$

Therefore the proposed finite difference method only requires one more forward passes to compute $\hat{\boldsymbol{f}}_{\phi_D}^{t-1}$ and instead, save 3 back-propagations. As back-propagation is typically more expensive than forward pass, our approach reduces running time drastically, especially when the sampler are applied to large neural network.

**Time complexity figures**    Every SG-MCMC method (including the meta sampler) requires $\nabla_{\boldsymbol{\theta}} \tilde{U}(\boldsymbol{\theta})$. The main burden is the forward pass and back-propagation through the $\boldsymbol{D}(\boldsymbol{z})$ and $\boldsymbol{Q}(\boldsymbol{z})$ matrices, where the latter one has been replaced by the proposed finite difference scheme. The time complexity is $O(HD)$ for both forward pass and finite difference with $H$ the number of hidden units in the neural network of the meta sampler. Parallel computation with GPUs improves real-time speed, indeed in our MNIST experiment the meta sampler spends roughly 1.5x time when compared with SGHMC.

## C    DETAILS OF THE STEIN GRADIENT ESTIMATOR

For a distribution $q(\boldsymbol{\theta})$ that is *implicitly* defined by a generative procedure, the density $q(\boldsymbol{\theta})$ is often intractable. Li & Turner (2018) derived the *Stein gradient estimator* that estimates $\mathbf{G} = (\nabla_{\boldsymbol{\theta}^1} \log q(\boldsymbol{\theta}^1), \cdots \nabla_{\boldsymbol{\theta}^K} \log q(\boldsymbol{\theta}^K))^{\mathrm{T}}$ on samples $\boldsymbol{\theta}^1, ..., \boldsymbol{\theta}^K \sim q(\boldsymbol{\theta})$. There are two different ways to derive this gradient estimator, here we briefly introduce one of them, and refer the readers to Li & Turner (2018) for details.

We start by introducing *Stein's identity* (Stein, 1972; 1981; Gorham & Mackey, 2015; Liu et al., 2016). Let $\boldsymbol{h} : \mathbb{R}^{d \times 1} \to \mathbb{R}^{d' \times 1}$ be a differentiable multivariate test function which maps $\boldsymbol{\theta}$ to a column vector $\boldsymbol{h}(\boldsymbol{\theta}) = [h_1(\boldsymbol{\theta}), h_2(\boldsymbol{\theta}), ..., h_{d'}(\boldsymbol{\theta})]^{\mathrm{T}}$. One can use *integration by parts* to show the following Stein's identity when a *boundary condition* $\lim_{||\boldsymbol{\theta}|| \to \infty} q(\boldsymbol{\theta})\boldsymbol{h}(\boldsymbol{\theta}) = 0$ is assumed for the test function:

$$\mathbb{E}_q[\boldsymbol{h}(\boldsymbol{\theta}) \nabla_{\boldsymbol{\theta}} \log q(\boldsymbol{\theta})^{\mathrm{T}} + \nabla_{\boldsymbol{\theta}} \boldsymbol{h}(\boldsymbol{\theta})] = \mathbf{0}, \quad \nabla_{\boldsymbol{\theta}} \boldsymbol{h}(\boldsymbol{\theta}) = (\nabla_{\boldsymbol{\theta}} h_1(\boldsymbol{\theta}), \cdots, \nabla_{\boldsymbol{\theta}} h_{d'}(\boldsymbol{\theta}))^{\mathrm{T}} \in \mathbb{R}^{d' \times d}. \tag{23}$$

This boundary condition holds for almost any test function if $q$ has sufficiently fast-decaying tails (e.g. Gaussian tails). Li & Turner (2018) proposed the *Stein gradient estimator* for $\nabla_{\boldsymbol{\theta}} \log q(\boldsymbol{\theta})$ by

inverting a Monte Carlo (MC) version of Stein's identity (23):

$$-\frac{1}{K}\mathbf{HG} \approx \overline{\nabla_{\boldsymbol{\theta}} \boldsymbol{h}}, \quad \mathbf{H} = \left(\boldsymbol{h}(\boldsymbol{\theta}^1), \cdots, \boldsymbol{h}(\boldsymbol{\theta}^K)\right) \in \mathbb{R}^{d' \times K}, \quad \overline{\nabla_{\boldsymbol{\theta}} \boldsymbol{h}} = \frac{1}{K}\sum_{k=1}^{K}\nabla_{\boldsymbol{\theta}^k}\boldsymbol{h}(\boldsymbol{\theta}^k) \in \mathbb{R}^{d' \times d}.$$

Then $\mathbf{G}$ is obtained by *ridge regression* (with $||\cdot||_F$ the Frobenius norm of a matrix)

$$\hat{\mathbf{G}}_V^{\text{Stein}} := \underset{\hat{\mathbf{G}} \in \mathbb{R}^{K \times d}}{\arg\min} ||\overline{\nabla_{\boldsymbol{\theta}} \boldsymbol{h}} + \frac{1}{K}\mathbf{H}\hat{\mathbf{G}}||_F^2 + \frac{\eta}{K^2}||\hat{\mathbf{G}}||_F^2, \quad \eta \geq 0, \tag{24}$$

which has an analytical solution

$$\hat{\mathbf{G}}_V^{\text{Stein}} = -(\mathbf{K} + \eta \boldsymbol{I})^{-1}\langle \nabla, \mathbf{K} \rangle, \tag{25}$$

where

$$\mathbf{K} := \mathbf{H}^{\mathrm{T}}\mathbf{H}, \quad \mathbf{K}_{ij} = \mathcal{K}(\boldsymbol{\theta}^i, \boldsymbol{\theta}^j) := \boldsymbol{h}(\boldsymbol{\theta}^i)^{\mathrm{T}}\boldsymbol{h}(\boldsymbol{\theta}^j),$$

$$\langle \nabla, \mathbf{K} \rangle := K\mathbf{H}^{\mathrm{T}}\overline{\nabla_{\boldsymbol{\theta}} \boldsymbol{h}}, \quad \langle \nabla, \mathbf{K} \rangle_{ij} = \sum_{k=1}^{K}\nabla_{\boldsymbol{\theta}^k(j)}\mathcal{K}(\boldsymbol{\theta}^i, \boldsymbol{\theta}^k).$$

Here $\boldsymbol{\theta}^k(j)$ denotes the $j^{\text{th}}$ element of vector $\boldsymbol{\theta}^k$. One can show that the RBF kernel satisfies Stein's identity (Liu et al., 2016). In this case $\boldsymbol{h}(\boldsymbol{\theta}) = \mathcal{K}(\boldsymbol{\theta}, \cdot), d' = +\infty$ and by the reproducing kernel property, $\boldsymbol{h}(\boldsymbol{\theta})^{\mathrm{T}}\boldsymbol{h}(\boldsymbol{\theta}') = \langle \mathcal{K}(\boldsymbol{\theta}, \cdot), \mathcal{K}(\boldsymbol{\theta}', \cdot)\rangle_{\mathcal{H}} = \mathcal{K}(\boldsymbol{\theta}, \boldsymbol{\theta}')$. Li & Turner (2018) also show that the Stein gradient estimator can be obtained by minimizing a Monte Carlo estimate of the kernelized Stein discrepancy (Chwialkowski et al., 2016; Liu et al., 2016).

**The kernel choice** It is well-known for kernel methods that a better choice of the kernel can greatly improve the performance. However, optimal kernels are often problem specific, and they are generally difficult to obtain. Recently, a popular approach for kernel design is to compose a simple kernel (e.g. RBF kernel) on features extracted from a deep neural network. Representative work include deep kernel learning for Gaussian processes (Wilson et al., 2016), and adversarial approaches to learn kernel parameters (Li et al., 2017; Bińkowski et al., 2018). Unfortunately, both approaches do not scale very well to our application as $\boldsymbol{\theta}$ has at least tens of thousands of dimensions. Furthermore, they both considered kernel learning for observed data, while in our case $\boldsymbol{\theta}$ is a latent variable to be inferred. Therefore it remains a research question on how to learn kernels on latent variables efficiently, and addressing this question is out of the scope of the paper. Instead, we follow Liu & Wang (2016); Li & Turner (2018) to use RBF kernel for the gradient estimator. Other kernels can be trivially adapted to our method. We expect even better performance if an optimal kernel is in use, but we leave the investigation to future work.

**Time complexity figures** During meta sampler training, the Stein gradient estimator requires the kernel matrix inversion which is $O(K^3)$ for cross-chain training. In practice, we only run a few parallel Markov chains $K = 20 \sim 50$, thus, this will not incur huge computation cost. For in-chain loss the computation can also be reduced with proper thinning schemes.

## D  IMPLEMENTATION DETAILS OF THE TRAINING LOSS

We visualize on the left panel of Figure 7 the unrolled computation scheme. We apply truncated back-propagate through time (BPTT) to train the sampler. Specifically, we manually stop the gradient flow through the input of $\boldsymbol{D}$ and $\boldsymbol{Q}$ matrices to avoid computing higher order gradients.

We also illustrate cross-chain in-chain training on the right panel of Figure 7. Cross-chain training encourages both fast convergence and low bias, provided that the samples are taken from parallel chains. On the other hand, in-chain training encourages sample diversity inside a chain. In practice, we might consider thinning the chains when performing in-chain training. Empirically this improves the Stein gradient estimator's accuracy as the samples are spread out. Computationally, this also prevents inverting big matrices for the Stein gradient estimator, and reduces the number of back-propagation operations. Another trick we applied is parallel chain sub-sampling: if all the chains are used, then there is less encouragement of singe chain mixing, since the parallel chain samples can be diverse enough already to give reasonable gradient estimate.

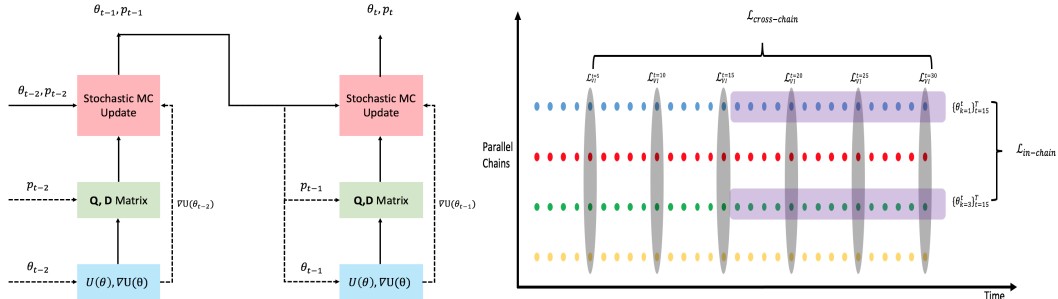

Figure 7: (Left) The unrolled scheme of the meta sampler updates. Stop gradient operations are applied to the *dashed* arrows. (Right) A visualization of cross-chain in-chain training. The grey area represents samples across multiple chains, and we compute the cross chain loss for every 5 time steps. The purple area indicates the samples taken across time with sub-sampled chains 1 and 3. In this visualization the initial 15 samples are discarded for burn-in, and the thinning length is $\tau = 1$ (effectively no thinning).

## E   INPUT PRE-PROCESSING

One potential challenge is that for different tasks and problem dimensions, the energy function, momentum and energy gradient can have very different scales and magnitudes. This affects the meta sampler's generalization, for example, if training and test densities have completely different energy scales, then the meta sampler is likely to produce wrong strategies. This is especially the case when the meta sampler is generalized to much bigger networks or to very different datasets.

To mediate this issue, we propose to pre-process the inputs to both $\boldsymbol{f}_{\phi_D}$ and $\boldsymbol{f}_{\phi_Q}$ networks to make it at similar scale as those in training task. Recall that the energy function is $U(\boldsymbol{\theta}) = -\sum_{n=1}^{N} \log p(\boldsymbol{y}_n|\boldsymbol{x}_n,\boldsymbol{\theta}) - \log p(\boldsymbol{\theta})$ where the prior $\log p(\boldsymbol{\theta})$ is often an isotropic Gaussian distribution. Thus the energy function scale linearly w.r.t both the dimensionality of $\boldsymbol{\theta}$ and the total number of observations $N$. Often the energy function is further approximated using mini-batches of $M$ datapoints. Putting them together, we propose pre-processing the energy as

$$\overline{U(\boldsymbol{\theta})} = \frac{1}{M}\sum_{m=1}^{M} \log p(\boldsymbol{y}_m|\boldsymbol{x}_m,\boldsymbol{\theta}) + \frac{D_{\text{train}}}{N D_{\text{test}}} \log p(\boldsymbol{\theta}) \tag{26}$$

where $D_{\text{train}}$ and $D_{\text{test}}$ are the dimensionality of $\boldsymbol{\theta}$ in the training task and the test task, respectively. Importantly, for RNNs $N$ represents the total sequence length, namely $N = \sum_{n=1}^{N_{data}} T_n$, where $N_{data}$ is the total number of sequences and $T_n$ is the sequence length for a datum $\boldsymbol{x}_n$. We also define $M$ accordingly. The momentum and energy gradient magnitudes are estimated by simulating a randomly initialized meta sampler for short iterations. With these statistics we normalize both the momentum and the energy gradient to have roughly zero mean and unit variance.

## F   EXPERIMENT SETUP

### F.1   TOY EXAMPLE

We train our meta sampler on a 10D uncorrelated Gaussian with mean $(3, ..., 3)$ and randomly generated covariance matrix. We do not set any offset and additional frictions, i.e. $\alpha = 0$ and $\beta = 0$. The noise estimation matrix $\tilde{B}$ are set to be 0 for both meta sampler and SGHMC. To mimic stochastic gradient, we manually inject Gaussian noise with zero mean and unit variance into $\nabla_{\boldsymbol{\theta}}\tilde{U}(\boldsymbol{\theta}) = \nabla_{\boldsymbol{\theta}}U(\boldsymbol{\theta}) + \boldsymbol{\epsilon}, \boldsymbol{\epsilon} \sim \mathcal{N}(\boldsymbol{0}, \boldsymbol{I})$. The functions $\boldsymbol{f}_{\phi_D}$ and $\boldsymbol{f}_{\phi_Q}$ are represented by 1-hidden-layer MLPs with 40 hidden units. For training task, the meta sampler step size is 0.01. The initial positions are drawn from $\text{Uniform}([0, 6]^D)$. We train our sampler for 100 epochs and each epochs consists 4 x 100 steps. For every 100 steps, we updates the $\boldsymbol{Q}$ and $\boldsymbol{D}$ matrices using Adam optimizer with learning rate 0.0005. Then we continue the updated sampler with last position and momentum until 4

sub-epochs are finished. We re-initialize the momentum and position. We use both cross-chain and in-chain losses. The Stein Gradient estimator uses RBF kernel with bandwidth chosen to be 0.5 times the median-heuristic estimated value. We unroll the Markov Chain for 20 steps before we manually stop the gradient. For cross-chain training, we take sampler across chain for each 2 time steps. For in-Chain, we discard initial 50 points for burn-in and sub-sample the chain with batch size 5. We thin the samples for every 3 steps. For both training and evaluation, we run 50 parallel Markov Chains.

The test task is to draw samples from a 20D correlated Gaussian with with mean $(3, ..., 3)$ and randomly generated covariance matrix. The step size is 0.025 for both meta sampler and SGHMC. To stabilize the meta sampler we also clamp the output values of $\boldsymbol{f}_{\phi_Q}$ within $[-5, 5]$. The friction matrix for SGHMC is selected as $\boldsymbol{I}$.

### F.2 BAYESIAN MLP MNIST

In MNIST experiment, we apply input pre-processing on energy function as in (26) and scale energy gradient by 70. Also, we scale up $\boldsymbol{f}_{\phi_D}$ by 50 to account for sum of stochastic noise. The offset $\alpha$ is selected as $\frac{0.01}{\eta}$ as suggested by Chen et al. (2014), where $\eta = \sqrt{\frac{lr}{N}}$ with $lr$ the per-batch learning rate. We also turn off the off-set and noise estimation, i.e. $\beta = 0$ and $\tilde{\boldsymbol{B}} = 0$. We run 20 parallel chains for both training and evaluation. We only adopt the cross chain training with thinning samplers of 5 times step. We also use the finite difference technique during evaluation to speed-up computations.

#### F.2.1 ARCHITECTURE GENERALIZATION (NT)

We train the meta sampler on a smaller BNN with architecture 784-20-10 and ReLU activation function, then test it on a larger one with architecture 784-40-40-10. In both cases the batch size is 500 following Chen et al. (2014). Both $\boldsymbol{f}_{\phi_D}$ and $\boldsymbol{f}_{\phi_Q}$ are parameterized by 1-hidden-layer MLPs with 10 units. The per-batch learning rate is 0.007. We train the sampler for 100 epochs and each one consists of 7 sub-epochs. For each sub-epoch, we run the sampler for 100 steps. We re-initialize $\boldsymbol{\theta}$ and momentum after each epoch. To stabilize the meta sampler in evaluation, we first run the meta sampler with small per-batch learning rate 0.0085 for 3 data epochs and clamp the $\boldsymbol{Q}$ values. After, we increase the per-batch learning rate to 0.018 with clipped $\boldsymbol{f}_{\phi_Q}$. The learning rate for SGHMC is 0.01 for all times. For SGLD and PSGLD, they are 0.2 and $1.4 \times 10^{-3}$ respectively. These step-sizes are tuned on MNIST validation data.

#### F.2.2 NT + ACTIVATION FUNCTION GENERALIZATION

We modify the test network's activation function to **sigmoid**. We use almost the same settings as in network generalization tests, except that the per-batch learning rates are tuned again on validation data. For the meta sampler and SGHMC, they are 0.18 and 0.15. For SGLD and PSGLD, they are 1 and $1.3 \times 10^{-2}$.

#### F.2.3 NT + DATASET GENERALIZATION

We train the meta sampler on ReLU network with architecture 784-20-5 to classify images 0-4, and test the sampler on ReLU network 784-40-40-5 to classify images 5-9. The settings are mostly the same as in network architecture generalization for both training and evaluation. One exception is again the per-batch learning rate for PSGLD, which is tuned as $1.3 \times 10^{-3}$. Note that even though we use the same per-batch learning rate as before, the discretization step-size is now different due to smaller training dataset, thus, $\alpha$ will be automatically adjusted accordingly.

### F.3 BAYESIAN CONVOLUTIONAL NEURAL NETWORK ON CIFAR-10

CIFAR-10 dataset contains 50,000 training images with 10 labels and 10,000 test images. We train our meta sampler using smaller CNN classifier with two convolutional layer ($3 \times 3 \times 3 \times 8$ and $3 \times 3 \times 8 \times 8$, no padding) and one fc layer of 50 hidden units. Therefore the dimensionality of $\boldsymbol{\theta}$ is $15,768$. The training sampler discretization step-size $\eta$ is $\sqrt{\frac{0.0007}{50000.}}$ and scaling term is $\alpha = \frac{0.005}{\eta}$. To make it analogous to optimization methods, we call 0.0007 as *per-batch learning rate* and 0.005

as *friction coefficient*. The $\boldsymbol{f}_{\phi_Q}$ and $\boldsymbol{f}_{\phi_D}$ are defined by 2-layer MLPs with 10 hidden units. We set the offset values to 0 for both $\boldsymbol{Q}$ and $\boldsymbol{D}$. Further, we scale up the output of $\boldsymbol{D}_f(\boldsymbol{z})$ by 10 and its gradient input $\nabla U(\boldsymbol{\theta})$ by 100. We scale up the energy input $U(\boldsymbol{\theta})$ to both $\boldsymbol{f}_{\phi_Q}$ and $\boldsymbol{f}_{\phi_D}$ by 5. We train our meta sampler using 100 "meta epoch" with 5 data epoch and 500 batch size. Within each "meta epoch", we repeat the following computation for 10 times: we run 50 parallel chains using the meta sampler for 50 iterations (0.5 dataset epoch), compute the loss function, and update the meta sampler's parameters using Adam. We manually stop the gradient after 20 iterations. Then we start the next sub-epoch using the last $\boldsymbol{\theta}$ and $\boldsymbol{p}$. After we finish all sub-epoch, we re-initialize the $\boldsymbol{\theta}$ and $\boldsymbol{p}$ using replay techniques with probability 0.15. The sub-sample chain number for in-chain loss is set to 5.

### F.3.1 THE REPLAY TECHNIQUE

Experience replay (Lin, 1993) is a technique broadly used in reinforcement learning literature. Inspired by this, in Bayesian CNN experiments we train the meta sampler in a similar way, and we found this replay technique particularly useful for more complicated dataset like CIFAR-10.

At the beginning of each "meta epoch", each chain is initialized either with a specific state randomly chosen from a replay pool, or with a random state sampled from a Gaussian distribution. We use a pre-defined replay probability to control the replay strategy. The replay pool is updated after each sub-epoch, and it has a queue-like data structure of constant size, so that the old states are replaced by the new ones. Therefore, this replay technique is useful for both short-time and long-time horizon generalization. On one hand, the meta sampler can continue with previous states, allowing it to accommodate long-time horizon behavior. On the other hand, due to non-zero probability of random restart, the meta sampler can learn a better strategy for fast convergence. Therefore with this replay technique, the sampler can observe both burn-in and roughly-converged behavior, and this balance is controlled by the replay probability.

### F.3.2 ARCHITECTURE GENERALIZATION (NT)

For architecture generalization, the test CNN has two convolutional layer ($3 \times 3 \times 3 \times 16$ and $3 \times 3 \times 16 \times 16$, no padding) and one fully connected layer with 100 hidden units. Thus, the dimensionality of $\boldsymbol{\theta}$ is 61,478, roughly 4 times of the training dimension. We run 20 parallel chains in test time. We split the 50,000 training images into 45,000 training and 5,000 validation images, and tune the discretization step-size of each sampling and optimization methods on the validation set for 80 epochs. For test, we run the tuned samplers/optimizers for 200 data epoch (roughly 40 times longer than training) to ensure convergence. For the meta sampler, the per-batch rate is 0.003. For SGHMC, the per-batch is also 0.003 with friction coefficient 0.01. For SGLD, the per-batch learning rate is 0.15. PSGLD uses $1.3 \times 10^{-3}$ as learning rate and 0.99 as moving average term. For optimization methods, we use learning rate 0.002 for Adam and 0.003 for SGD-M. The momentum term is 0.9. To prevent overfitting, we use weight penalty with coefficient 0.001.

### F.3.3 NT + SIGMOID GENERALIZATION

The test CNN has same architecture as in NT, except that it replaces all ReLU activation functions with sigmoid activations. We fix all other parameters for sampling method and only re-tune the step sizes using same setup as in NT. The per-batch rate for meta sampler, SGHMC, SGLD and PSGLD are 0.1, 0.03, 0.5 and 0.005 respectively. For optimization methods, the step size for Adam and SGD-M are 0.002 and 0.03 respectively.

### F.3.4 NT + DATASET GENERALIZATION

We split the CIFAR-10 training and test dataset according to the labels. We use training data with labels 0-4 for meta sampler training, training data with labels 5-9 for test CNN training, and test data with labels 5-9 for test CNN evaluation. Thus, the meta sampler has no access to the test task's training and test data during sampler training. We train our sampler using the same scaling terms as in NT but reduce the discretization step-size to 0.0005. The rest setup is the same as in NT.

We use the same test CNN architecture and ReLU activation as in NT, and tune the learning rate using validation data. The step size for the meta sampler, SGHMC, SGLD and PSGLD are 0.0015, 0.005,

Table 3: The basic statistics for 4 RNN datasets, bold figure represents large difference compared to others. *Size* is the number of data point. *Avg. Time* is the averaged sequence and *Energy scale* is the rough scale of the train NLL when sampler converges.

|  | Piano | Muse | Nott | JSB |
|---|---|---|---|---|
| Size:train | **87** | 524 | 694 | 229 |
| Size:test | 25 | 124 | 170 | 77 |
| Avg. Time:train | 872 | 467 | 254 | **60** |
| Avg. Time:test | 761 | 518 | 261 | **61** |
| Energy scale:train | $\approx 7.2$ | $\approx 7$ | $\approx \mathbf{2.5}$ | $\approx 7.8$ |

0.2 and 0.0018, respectively. For optimization methods, we use learning rates 0.002 and 0.003 for Adam and SGD-M respectively.

## F.4    BAYESIAN RNN

The *Piano* data is selected as the training task, which is further split into training, validation and test subsets. We use batch-size 1, meaning that the energy and the gradient are estimated on a single sequence. The meta sampler uses similar neural network architectures as in MNIST tests. The training and evaluation per-batch learning rate for all the samplers is set to be 0.001 following Chen et al. (2016). We train the meta sampler for 40 epochs with 7 sub-epochs with only cross chain loss. Each sub-epochs consists 70 iterations. We scale the $\boldsymbol{D}$ output by 20 and set $\alpha = \frac{0.002}{\eta}$, where $\eta$ is defined in the same way as before. We use zero offset during training, i.e. $\beta = 0$. We apply input pre-processing for both $\boldsymbol{f}_{\phi_D}$ and $\boldsymbol{f}_{\phi_Q}$. To prevent divergence of the meta sampler at early training stage. We also set the constant of $c = 100$ to the $f_{\phi_D}$. For dataset generalization, we tune the off-set value based on Piano validation set and transfer the tuned setting $\beta = -1.5$ to the other three datasets. For Piano architecture generalization, we do not tune any hyper-parameters including $\beta$ and use exactly same settings as training. Exact gradient is used in RNN experiments instead of computing finite differences.

## G    RNN DATASET DESCRIPTION

We list some data statistics in Table 3 which roughly indicates the similarity between datasets. Piano dataset is the smallest in terms of data number, however, the averaged sequence length is the largest. Muse dataset is similar to Piano in sequence length and energy scale but much larger in terms of data number. On the other hand, Nott dataset has very different energy scale compared to the other three. This potentially makes the generalization much harder due to inconsistent energy scale fed into $\boldsymbol{f}_{\phi_Q}$ and $\boldsymbol{f}_{\phi_D}$. For JSB, we notice a very short sequence length on average, therefore the GRU model is more likely to over-fit. Indeed, some algorithms exhibits significant over-fitting behavior on JSB dataset compared to other data (Santa is particularly severe).

## H    ADDITIONAL PLOTS

### H.1    SHORT HORIZON PERFORMANCE COMPARISONS

We also run the samplers using the same settings as in MNIST experiments for a short period of time (500 iterations). We also compare to other optimization methods including momentum SGD (SGD-M) and Adam. We use the same per-batch learning rate for SGD-M and SGHMC as in MNIST experiment. For Adam, we use 0.002 for ReLU and 0.01 for Sigmoid network.

The results are shown in Figure 8. Meta sampler and Adam achieves the fastest convergence speed. This again confirms the faster convergence of the meta sampler especially at initial stages. We also provide additional contour plots (Figure 9) for MNIST experiments to demonstrate the strategy learned by $\boldsymbol{f}_{\phi_D}$ for reference.

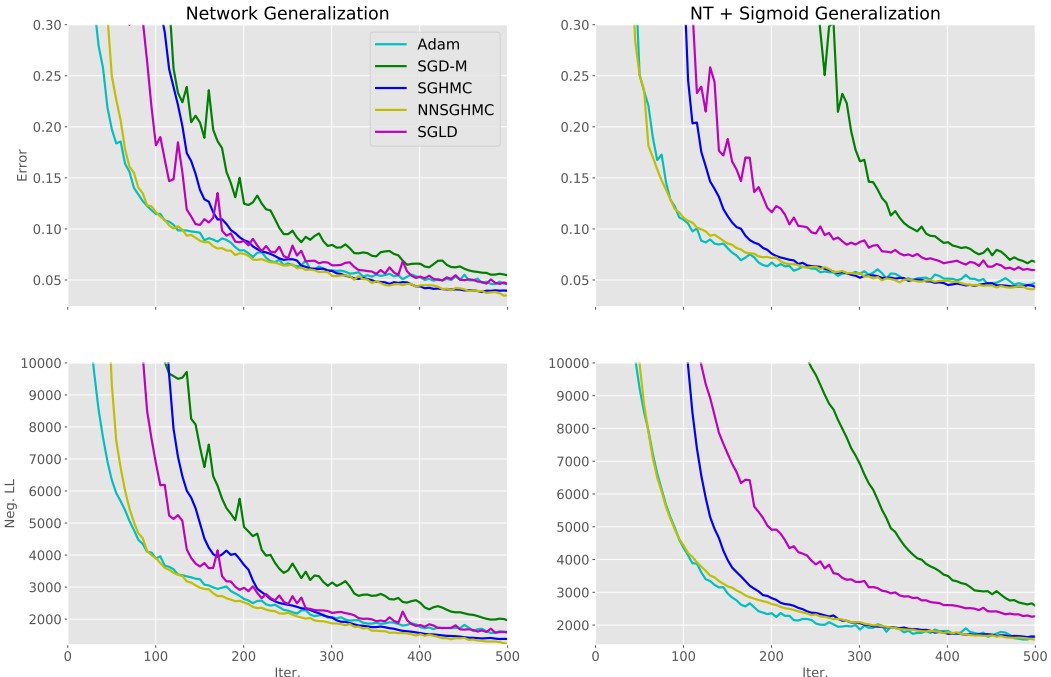

Figure 8: We only test the *Network Generalization* and *Activation function generalization*. The **upper** part indicates the test error plot and **lower** part are the negative test LL curve

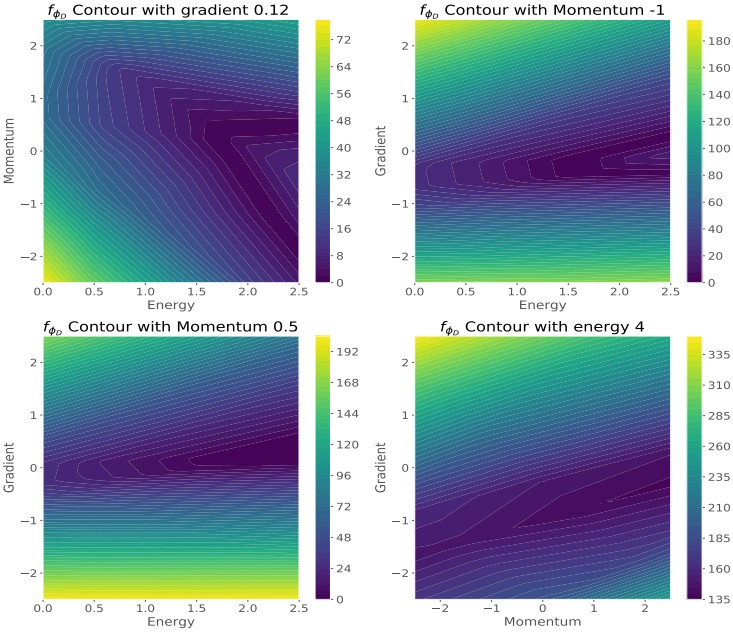

Figure 9: The contour plots of $f_{\phi_D}$ for other input values.

