# OpenReview forum: "Meta-Learning For Stochastic Gradient MCMC"
_ICLR.cc/2019/Conference_

### Official Review · AnonReviewer2 · 2018-11-02
**Very well written paper, interesting idea, unconvincing experimental results**

**Rating:** 6
**Confidence:** 3

**Review:**

This paper proposes a novel method to perform meta-learning for stochastic gradient MCMC. They utilize a general family of SDEs that guarantees preservation of the target density with somewhat loose constraint on the drift and diffusion functions (from Ma et al. (2015)). Then, they propose learning these functions on a set of training tasks and evaluating on unseen, different tasks, in a meta-learning fashion.

This paper is well written and easy to follow. They do a very good job presenting the motivation for their work as well as seminal work in SG-MCMC. The idea is fairly natural, especially in light of recent success of meta-learning and learning optimizers. They do a thorough survey of related work and also do a good job presenting their method in context of very modern work on MCMC and SG-MCMC.

I am not completely convinced by the meta-training objective; both losses seem natural but quite intractable to compute in practice. The use of Stein indicates that the kernel must probably be *very* carefully crafted and given that the whole method relies on this objective, it seems like this could be a breaking point. I am also curious to know how you diagnostic/evaluate the choice of these kernels.

In terms of evaluation, the experimental results are not the most convincing given that across the board, they are (except in one case) in 4 case within 0.2% of SGHMC and in the two others, within 0.5% and 0.8% respectively. This seems a bit weak, especially considering the compute invested both at training time and for each SG-MCMC step (i.e. getting the outputs from the neural networks vs simply doing HMC). Is there really a case for using the method over SG-HMC? I would have also very much liked to see a run-time evaluation.

---

> ### Author Response · Authors · 2018-11-14
> **Reply from Author**
>
> We thank the reviewer for his/her valuable time for detailed reviews on our submission. Here are our responses for the concerns raised by Reviewer 2:
>
> Q1: Concerns about Stein gradient estimator.
>
> A1: The main objective of this paper is to propose a meta-learning algorithm for SG-MCMC, and the investigation for better kernels is not our main focus. We used RBF kernel in our experiments, and empirically we didn’t find the objective to be very sensitive to the hyper-parameters of this kernel choice.
>
> We expect a better kernel choice would improve the performance of the meta-sampler even further. However, the choice of kernel for Stein discrepancy remains an open challenge. The typical choices are the RBF kernel with median heuristics [Liu et al, 2016, Chwialkowski et al, 2016, Li & Turner, 2018; Shi et al., 2018] or the IMQ kernel [Gorham & Mackey, 2017, Chen, et al., 2018; Li, 2017]. In this paper, we adopt the settings of RBF kernel, but other kernels can be easily applied. We thank you for pointing out this and we will add discussions in revision.
>
> Q2: Weak results
>
> A2: Our results indeed show significant improvement when compared with SG-MCMC literature. E.g. on MNIST, even with much bigger neural nets, hand-designed SG-MCMC samplers often show around 0.2% improvements over baselines, see Figure 4 in [Chen, et al., 2014] and table 2 in [ Li, et al, 2016].
>
> For Cifar-10 experiments, we use the same settings as [Luo, et al., 2017] which is the current state-of-the-art SG-MCMC sampler. Their approach improved performance over SGHMC by 1.3% (over our SGHMC baseline), and our approach improved over SGHMC by 0.5%. However, their method uses 5 augmented variables with very carefully engineered dynamics design, while our approach only uses 1 augmented variable (momentum), and the dynamics are learned from data. Thus, we argue that by using the same augmentation as SGHMC, this performance increase is significant. The meta-learning idea is also applicable to improve the sampler of [Luo et al. 2017] which we leave to future work.
>
> Q3: Computation costs.
>
> A3: We report the plots in terms of epochs, making the metric consistent with other SG-MCMC papers. For wall clock time concern, in MNIST experiments the meta sampler took around 1.3-1.5x time when compared with SGHMC (see the last paragraph in appendix B).
>
> As an important side note, to the best of our knowledge, none of the existing meta-learning optimisation papers has reported results in wall clock time. We presume these meta-learned optimisers can be much slower than Adam in real time, however this cost can be amortised using parallel computing. Our proposed sampler can be easily parallelized too, and in such case convergence speed in terms of number of iterations/epochs is more important.
>
> Reference:
>
> Chen et al. 2016. "Stochastic gradient hamiltonian monte carlo." ICML. 2014.
>
> Li, et al. 2016. "Preconditioned Stochastic Gradient Langevin Dynamics for Deep Neural Networks." AAAI 2016.
>
> Liu et al, 2016. “A Kernelized Stein Discrepancy for Goodness-of-fit Tests and Model Evaluation”. ICML 2016.
>
> Chwialkowski et al, 2016. “A kernel test of goodness of fit”. ICML 2016.
>
> Liu & Wang, 2016. “Stein Variational Gradient Descent: A General Purpose Bayesian Inference Algorithm”. NIPS 2016.
>
> Gorham & Mackey, 2017. “Measuring Sample Quality with Kernels”. ICML 2017
>
> Li, 2017. "Approximate Gradient Descent for Training Implicit Generative Models." NIPS 2017 Bayesian deep learning workshop.
>
> Luo, et al. 2017. "Thermostat-assisted Continuous-tempered Hamiltonian Monte Carlo for Multimodal Posterior Sampling on Large Datasets." arXiv preprint arXiv:1711.11511 .
>
> Li & Turner, 2018. "Gradient estimators for implicit models." ICLR 2018.
>
> Shi et al., 2018. "A Spectral Approach to Gradient Estimation for Implicit Distributions." ICML 2018.
>
> Chen et al. 2018. "Stein points." ICML 2018.

---

> > ### Comment · AnonReviewer2 · 2018-11-26
> > **Response**
> >
> > I have read your response and thank you for your clarifications.
> >
> > Thank you for the precision regarding Stein's; I do agree it is not a significant part of your work but I was wondering if this could be a particularly fragile one. The fact that it does not seem sensitive to kernel choice/hyperparameters seem to be a good indication.
> >
> > I do not change my initial assessment of the paper as I still think the empirical gains are not very convincing; especially in light of a 1.3-1.5x slowdown.

---

### Official Review · AnonReviewer3 · 2018-11-02
**Very interesting paper but many design choices not validated**

**Rating:** 7
**Confidence:** 4

**Review:**

TITLE
Meta-learning for stochastic gradient mcmc

REVIEW SUMMARY
A wonderful paper with many great ideas and insights. Main weakness is the complexity of the algorithms and many design choices wich are well argued for but not theoretically or empirically well founded.

PAPER SUMMARY
The main idea (based on the result of Ma et al.'s "complete recipe for stochastic gradient mcmc") is to parameterize the diffusion and curl matrices by neural networks and (meta-)learn/optimize an sg-mcmc algorithm.

QUALITY
The technical quality of the paper appears to be good. Due to the complexity of the algorithm and lack of access to authors code at review time, it is not feasible for me to validate empirical results.

My main critisism of this work is that the proposed procedure is quite complicated, and there are a lot of steps and design choices that are made in the paper which are not backed up by theory or experiment. For example, the structure and parametrization of D and Q. I would like to have seen e.g. empirical results on full matrices compared to the particular "diagonal" struture used, to give an idea of how much we loose by that design choise. Similarly, the choice of meta learning objective is not (to me at least) obvious, and this could be examined further. Also, the use of the Stein gradient estimator is known sometimes to be problematic (maybe particularly with an rbf kernel) but this is not explored.

All in all, the paper leaves me wanting more, but of course there is only so much space in a conference paper. My conclusion here is that I recommend that the paper is published as it is, and I hope the authors will continue their work in future research (as also outlined in the paper).

CLARITY
The paper is clear and well written, notation is consistent, and everything is fairly easy to follow.

ORIGINALITY
The idea of meta learning sg-mcmc is not something I have seen before, so to my knowledge the idea is original.

SIGNIFICANCE
I think the whole line of research in which this paper falls has a very high potential, and i strongly welcome any new results. This paper develops new interesting ideas of broad interest.

---

> ### Author Response · Authors · 2018-11-14
> **Reply from Author**
>
> We thank the reviewer for his/her valuable time for detailed reviews on our submission. Here are our responses for the concerns raised by Reviewer 3:
>
> Q1: “the proposed procedure is quite complicated”
>
> A1:  In theory the designing choice of D and Q can be arbitrary and the resulting sampler is still valid due to the completeness results of Ma et al. In practice, however, there are several constraints when concerning scalability. Here \theta has dimensionality around 15,000 even for the smallest MLP we tested (1 hidden layer with 20 hidden units). Therefore, after momentum variable augmentation, if full rank matrices are used then the Q and D matrices will have around 30000^2 (O(d^2)) entries. Furthermore, computing D^{1/2} has O(d^3) cost, in this example it would be 30000^3. In sum these high costs make full rank matrix design prohibitive, and we resort to diagonal matrices for better scalability.
>
> Q2: “choice of meta learning objective is not obvious”
>
> A2: The proposed two losses target different aspects of the MCMC chain. The cross-chain objective encourages q_t to approach the exact posterior faster. By definition, if q_t = p then the sampler has converged to the stationary distribution. However, cross-chain loss does not reflect mixing properties within a single chain, even when the chain is initialised by samples from p. Therefore we developed the in-chain loss to improve single chain mixing, which encourages the underlying distribution of in-chain samples to have high entropy.
>
> The choice of divergence in use is a bit tricky. First in MCMC the q distribution is implicitly defined via parallel chain simulation and/or thinning. On the other hand, we can only evaluate the exact posterior (up to a constant) at a given input \theta. Furthermore evaluating p on the full dataset can be costly, what we have in practice is an unbiased estimate of log p(\theta|D) with mini-batches. These two observations motivate the usage of KL[q||p], where the intractable gradient problem for H[q] is addressed by the Stein gradient estimator (see answers to the next question).
>
> We do not use GAN-based approaches (as done in some of the citations) since in this case we do not have “real data” from the exact posterior, and \theta is at least 15,000 dimensions in our smallest MLP. As for integral probability metrics, the only possible choice is kernelized Stein discrepancy (KSD) [Liu et al, 2016, Chwialkowski et al, 2016]. But as shown in [Liu & Wang, 2016], minimising KSD is equivalent to minimising the norm of the gradient of KL in a unit ball of an RKHS. This means the two divergences are closely related, applications of KSD to our task can be an interesting future direction.
>
> Q3: “the use of the Stein gradient estimator is known sometimes to be problematic”
>
> A3: The main objective of this paper is to propose a meta-learning algorithm for SG-MCMC, and the investigation for better kernels is not our main focus. We used RBF kernel in our experiments, and empirically we didn’t find the objective to be very sensitive to the hyper-parameters of this kernel choice.
>
> We expect a better kernel choice would improve the performance of the meta-sampler even further. However, the choice of kernel for Stein discrepancy remains an open challenge. The typical choices are the RBF kernel with median heuristics [Liu et al, 2016, Chwialkowski et al, 2016, Li & Turner, 2018; Shi et al., 2018] or the IMQ kernel [Gorham & Mackey, 2017, Chen, et al., 2018; Li, 2017]. In this paper, we adopt the settings of RBF kernel, but other kernels can be easily applied. We thank you for pointing out this and we will add discussions in revision.
>
> Reference:
>
> Liu et al, 2016. “A Kernelized Stein Discrepancy for Goodness-of-fit Tests and Model Evaluation”. ICML 2016.
>
> Chwialkowski et al, 2016. “A kernel test of goodness of fit”. ICML 2016.
>
> Liu & Wang, 2016. “Stein Variational Gradient Descent: A General Purpose Bayesian Inference Algorithm”. NIPS 2016.
>
> Gorham & Mackey, 2017. “Measuring Sample Quality with Kernels”. ICML 2017
>
> Li, 2017. "Approximate Gradient Descent for Training Implicit Generative Models." NIPS 2017 Bayesian deep learning workshop.
>
> Li & Turner, 2018. "Gradient estimators for implicit models." ICLR 2018.
>
> Shi et al., 2018. "A Spectral Approach to Gradient Estimation for Implicit Distributions." ICML 2018.
>
> Chen et al. 2018. "Stein points." ICML 2018.

---

> > ### Comment · AnonReviewer3 · 2018-12-04
> > **Thank you**
> >
> > Thank you very much for the elaboration and clarification

---

### Official Review · AnonReviewer1 · 2018-11-02
**Well written paper that presents an interesting proof-of-concept for meta-learning MCMC samplers**

**Rating:** 7
**Confidence:** 4

**Review:**

In the paper "Meta-Learning for Stochastic Gradient MCMC", the authors present a meta-learning approach to automatically design MCMC sampler based on Hamiltonian dynamics to mix faster on problems similar to the training problems. The approach is evaluated on simple multidimensional Gaussians, and Bayesian neural networks (including fully connected, convolutional, and recurrent networks).

MCMC samplers in general, and Hamiltonian Monte Carlo sampler in particular, are very powerful tools to perform Bayesian inference in high-dimensional spaces. Combined with stochastic gradients, methods like Stochastic Gradient MCMC (SGMCMC), or Stochastic Gradient Langevin Dynamics (SGLD) have been successfully used to apply these methods in the large data regime, where only noisy estimates of the gradients are feasible. Even though, many different samplers exists, and they are provably correct (meaning they converge to the correct distribution), fast mixing and low auto-correlation within the chain can heavily depend on the problem at hand and the hyperparameters of the sampler used.  The work presented here, uses the general framework for SG-MCMC samplers of Ma et al., parametrizes it with a neural network and learns its weights on representative training problems.

The paper is well written, although occasional minor mistakes and typos can be found.
It seems however, that the method is still quite laborious and some care needs to be taken to train the meta-sampler.
The overall narrative is easy to follow, but could benefit from more detail in certain parts. In general, I argue for acceptance of the paper, but have the following questions/comments:

- Below Eq. (7), an interpretation of the parametrizations Q_f and D_f is given. I greatly appreciate this, but the phrase 'Q_f is
responsible for the acceleration of \theta' is not really instructive. By definition, the change in \theta is mostly driven by the momentum p. Therefor, Q_f looks like an inverse mass (at least in the second line of (7)), but maybe that is not a very helpful analogy either.
- at the beginning of section 3.2, the term 'particles' is used. While I am fully aware of what that is supposed to mean, a reader less familiar with the topic could be confused, because there is no explanation of it.
- It is unclear to me how the stochastic estimate \tilde{U}(\theta) in equation (10) is computed exactly. Is it estimated using the current mini-batch at time t, or is it estimated using a 'holdout-test set'?
- I was wondering how the correlation between the chains due to thinning for the In-chain loss affects the results. The text, does not address this at all.
- The experiments are very thorough and I appreciate the comparison to the tuned baselines, but I am missing some details in the paper:
     (a) Did you tune the SGHMC method in Figure 2, as well? It is not mentioned in the text, and the sample path looks very volatile, which could indicate a poor combination of step length and noise.
     (b) How was the tuning of the base line methods performed?
     (c) Are the results in Figure 3 based on single runs, or do you show the mean over 10 independent runs (as in table 1).
- The insets in Figure 6 are helpful, but I think you could shrink the 'outer y axis' and have the inset in the top right corner instead. That way, the zoomed-out plot would show more details on its own.

---

> ### Author Response · Authors · 2018-11-14
> **Reply from Author**
>
> We thank the reviewer for his/her valuable time for detailed reviews on our submission. Here are our responses for the concerns raised by Reviewer 1:
>
> Q1:  “the phrase 'Q_f is responsible for the acceleration of \theta' is not really instructive”
>
> A1: The Q_f value should not be viewed as the inverse mass matrix, otherwise Q_f should appear at the kinetic energy term inside Hamiltonian term H, where the kinetic term will be p^T Q_f p / 2. The resulting update rule will be different from Eq. 7. Additionally, the Q matrix should satisfy the anti-symmetry property which is not required for inverse mass matrix.
> In fact, from the first line in Eq.7, we noticed that Q_f is also responsible for scaling of driven forces for momentum p (appears at the \nabla_{\pmb{\theta}}\pmb{\tilde{U}}), similarly as in the second line of Eq.7. Thus, we think Q_f controls both the scaling of momentum p and \pmb{\theta}. Therefore, we conclude it as the acceleration force for \pmb{\theta}
>
> Q2: “how the stochastic estimate \tilde{U}(\theta) in equation (10) is computed”
>
> A2: The energy function is estimated using the current mini-batch training data. To be specific, at the end of time t, we have a set of K \theta samples from K parallel chains, and mini-batch observed data with batch size M drawn from the training data set, we use these mini-batch data to estimate U using Eq. 4 for each \theta_k.
>
> Q3: ”how the correlation between the chains due to thinning for the in-chain loss affects the results”
>
> A3: The chains are run completely in parallel, thus there is no correlation between chains. As for the entropy term in in-chain loss, the gradient estimator took samples inside each chain, and the back-propagation is through each single chain.
>
> Q4:  “Did you tune the SGHMC method in Figure 2, as well?”
>
> A4: We use the same step size for both SGHMC and our meta sampler, and that step size is set to be very small in order to reflect the behaviour of the continuous dynamics. We agree that SGHMC with carefully tuned hyper-parameters can perform well, however the point of meta learning is exactly to avoid laborious hyper-parameter tuning by humans, and instead to learn them automatically from data. Indeed this advantage is clearly demonstrated by this synthetic example.
>
> Q6:  “How was the tuning of the baseline methods performed?”
>
> A6: To test a meta-learning algorithm, we need to define both the training task and the test taks. Each task contains its own training/validation/test datasets. We train the meta-sampler on the training task. For evaluation, we tune the hyper-parameters of both the baseline samplers and the meta-sampler on the **validation set** of the test task, and report the performances on the **test set** of the test task.
>
> Q7: “Are the results in Figure 3 based on single runs?”
>
> A7: Results in Figure 3 shows both the mean and standard error over 10 runs as in table 1. However the standard errors are too small compared to the magnitude of the classifier error, so it may be not so clear in the figure 3. Will improve this on revision.

---

### Author Response · Authors · 2018-11-25
**Revision available**

Thank you for your reviews, we have revised our submission based on some of the reviews below:

1. We add some comments on using RBF kernel for Stein gradient estimator in the Appendix C, explaining some of the concerns raised by the reviewers.

2. We re-plot the Figure 6

3. 'Particles' are now replaced by 'Samples' to avoid any confusions.

Best,

Paper383 Author

---

### Meta-Review · Area_Chair1 · 2018-12-14

**Confidence:** 4
**Recommendation:** Accept (Poster)

**Metareview:**

This paper proposes to use meta-learning to design MCMC sampling distributions based on Hamiltonian dynamics, aiming to mix faster on set of problems that are related to the training problems. The reviewers agree that the paper is well-written and the ideas are interesting and novel. The main weaknesses of the paper are that (1) there is not a clear case for using this method over SG-HMC, and (2) there are many design choices that are not validated. The authors revised the paper to address some aspects of the latter concern, but are encouraged to add additional revisions to clarify the points brought up by the reviewers.
Despite the weaknesses, the reviewers all agree that the paper exceeds the bar for acceptance. I also recommend accept.